# Federated Virtual Learning on Heterogeneous Data with Local-global Distillation

## Abstract

Despite Federated Learning (FL)'s trend for learning machine learning models in a distributed manner, it is susceptible to performance drops when training on heterogeneous data. In addition, FL inevitability faces the challenges of synchronization, efficiency, and privacy. Recently, dataset distillation has been explored in order to improve the efficiency and scalability of FL by creating a smaller, synthetic dataset that retains the performance of a model trained on the local private datasets. *We discover that using distilled local datasets can amplify the heterogeneity issue in FL.* To address this, we propose a new method, called **Fed**erated Virtual Learning on Heterogeneous Data with **L**ocal-**G**lobal **D**istillation (FEDLGD), which trains FL using a smaller synthetic dataset (referred as *virtual data*) created through a combination of local and global dataset distillation. Specifically, to handle synchronization and class imbalance, we propose iterative distribution matching to allow clients to have the same amount of balanced *local virtual data*; to harmonize the domain shifts, we use federated gradient matching to distill *global virtual data* that are shared with clients without hindering data privacy to rectify heterogeneous local training via enforcing local-global feature similarity. We experiment on both benchmark and real-world datasets that contain heterogeneous data from different sources, and further scale up to an FL scenario that contains large number of clients with heterogeneous and class imbalance data. Our method outperforms *state-of-the-art* heterogeneous FL algorithms under various settings with a very limited amount of distilled virtual data.

## 1 Introduction

Federated Learning (FL) [29] has become a popular solution for different institutions to collaboratively train machine learning models without pooling private data together. Typically, it involves a central server and multiple local clients; then the model is trained via aggregation of local network parameter updates on the server side iteratively. FL is widely accepted in many areas, such as computer vision, natural language processing, and medical image analysis [25, 12, 41].

On the one hand, clients with different amounts of data cause asynchronization and affect the efficiency of FL systems. Dataset distillation [39, 5, 46, 44, 45] addresses the issue by only summarizing smaller synthetic datasets from the private local datasets to ensure each client owns the same amount of data. We refer this underexplored strategy as *federated virtual learning*, as the models are trained from synthetic data [40, 10, 16]. These methods have been found to perform better than model-synchronization-based FL approaches while requiring fewer server-client interactions.

On the other hand, due to different data collection protocols, data from different clients inevitably face heterogeneity problems with domain shift, which means data may not be independent and identically distributed (iid) among clients. Heterogeneous data distribution among clients becomes a

key challenge in FL, as aggregating model parameters from non-iid feature distributions suffers from client drift [18] and diverges the global model update[26].

We observe that using locally distilled datasets can amplify the heterogeneity issue. Figure 1 shows the tSNE plots of two different datasets, USPS [31] and SynthDigits [9], each considered as a client. tSNE takes the original and distilled virtual images as input and embeds them into 2D planes. One can observe that the distribution becomes diverse after distillation.

To alleviate the problem of data heterogeneity in classical FL settings, two main orthogonal approaches can be taken. *Approach 1* aims to minimize the difference between the local and global model parameters to improve convergence [25, 18, 38]. *Approach 2* enforces consistency in local embedded features using anchors and regularization loss [37, 47, 42]. The first approach can be easily applied to distilled local datasets, while the second approach has limitations when adapting to federated virtual learning. Specifically, VHL [37] samples global anchors from untrained StyleGAN [19] suffers performance drop when handling amplified heterogeneity after dataset distillation. Other methods, such as those that rely on external global data [47], or feature sharing from clients [42],

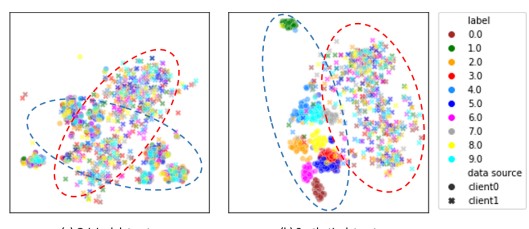

(a) Original datasets      (b) Synthetic datasets

Figure 1: Distilled local datasets can worsen heterogeneity in FL. tSNE plots of (a) original datasets and (b) distilled virtual datasets of USPS (client 0) and SynthDigits (client 1). The two distributions are marked in red and blue. We observe fewer overlapped ○ and × in (b) compared with (a), indicating higher heterogeneity between two clients after distillation.

are less practical, as they pose greater data privacy risks compared to classical FL settings[1]. *Without hindering data privacy,* developing strategies following *approach 2* for federated virtual learning on heterogeneous data remains open questions on *1) how to set up global anchors for locally distilled datasets and 2) how to select the proper regularization loss(es).*

To this end, we propose FEDLGD, a federated virtual learning method with local and global distillation. We propose *iterative distribution matching* in local distillation by comparing the feature distribution of real and synthetic data using an evolving feature extractor. The local distillation results in smaller sets with balanced class distributions, achieving efficiency and synchronization while avoiding class imbalance. FEDLGD updates the local model on local distilled synthetic datasets (named *local virtual data*). We found that training FL with local virtual data can exacerbate heterogeneity in feature space if clients' data has domain shift (Figure. 1). Therefore, unlike previously proposed federated virtual learning methods that rely solely on local distillation [10, 40, 16], we also propose a novel and efficient method, *federated gradient matching*, that integrated well with FL to distill global virtual data as anchors on the server side. This approach aims to alleviate domain shifts among clients by promoting similarity between local and global features. Note that we only share local model parameters w.r.t. distilled data. Thus, the privacy of local original data is preserved. We conclude our contributions as follows:

- This paper focuses on an important but underexplored FL setting in which local models are trained on small distilled datasets, which we refer to as *federated virtual learning*. We design two effective and efficient dataset distillation methods for FL.

- We are *the first* to reveal that when datasets are distilled from clients' data with domain shift, the heterogeneity problem can be *exacerbated* in the federated virtual learning setting.

- We propose to address the heterogeneity problem by mapping clients to similar features regularized by gradually updated global virtual data using averaged client gradients.

- Through comprehensive experiments on benchmark and real-world datasets, we show that FEDLGD outperforms existing state-of-the-art FL algorithms.

---

[1]Note that FedFA [47], and FedFM [42] are unpublished works proposed concurrently with our work

## 2 Related Work

### 2.1 Dataset Distillation

Data distillation aims to improve data efficiency by distilling the most essential feature in a large-scale dataset (e.g., datasets comprising billions of data points) into a certain terse and high-fidelity dataset. For example, Gradient Matching [46] is proposed to make the deep neural network produce similar gradients for both the terse synthetic images and the original large-scale dataset. Besides, [5] proposes matching the model training trajectory between real and synthetic data to guide the update for distillation. Another popular way of conducting data distillation is through Distribution Matching [45]. This strategy instead, attempts to match the distribution of the smaller synthetic dataset with the original large-scale dataset. It significantly improves the distillation efficiency. Moreover, recent studies have justified that data distillation also preserves privacy [7, 4], which is critical in federated learning. In practice, dataset distillation is used in healthcare for medical data sharing for privacy protection [22]. Other modern data distillation strategies can be found here [33].

### 2.2 Heterogeneous Federated Learning

FL performance downgrading on non-iid data is a critical challenge. A variety of FL algorithms have been proposed ranging from global aggregation to local optimization to handle this heterogeneous issue. *Global aggregation* improves the global model exchange process for better unitizing the updated client models to create a powerful server model. FedNova [38] notices an imbalance among different local models caused by different levels of training stage (e.g., certain clients train more epochs than others) and tackles such imbalance by normalizing and scaling the local updates accordingly. Meanwhile, FedAvgM [15] applies the momentum to server model aggregation to stabilize the optimization. Furthermore, there are strategies to refine the server model from learning client models such as FedDF [27] and FedFTG [43]. *Local training optimization* aims to explore the local objective to tackle the non-iid issue in FL system. FedProx [25] straightly adds $L_2$ norm to regularize the client model and previous server model. Scaffold [18] adds the variance reduction term to mitigate the "clients-drift". Also, MOON [24] brings mode-level contrastive learning to maximize the similarity between model representations to stable the local training. There is another line of works [42, 37] proposed to use a global *anchor* to regularize local training. Global anchor can be either a set of virtual global data or global virtual representations in feature space. However, in [37], the empirical global anchor selection may not be suitable for data from every distribution as they don't update the anchor according to the training datasets.

### 2.3 Datasets Distillation for FL

Dataset distillation for FL is an emerging topic that has attracted attention due to its benefit for efficient FL systems. It trains model on distilled synthetic datasets, thus we refer it as federated virtual learning. It can help with FL synchronization and improve training efficiency by condensing every client's data into a small set. To the best of our knowledge, there are few published works on distillation in FL. Concurrently with our work, some studies [10, 40, 16] distill datasets locally and share the distilled datasets with other clients/servers. Although privacy is protected against *currently* existing attack models, we consider sharing local distilled data a dangerous move. Furthermore, none of the existing work has addressed the heterogeneity issue.

## 3 Method

In this section, we will describe the problem setup, introduce the key technical contributions and rationale of the design for FEDLGD, and explain the overall training pipeline.

### 3.1 Setup for Federated Virtual Learning

We start with describing the classical FL setting. Suppose there are $N$ parties who own local datasets $(D_1, \ldots, D_N)$, and the goal of a classical FL system, such as FedAvg [29], is to train a global model

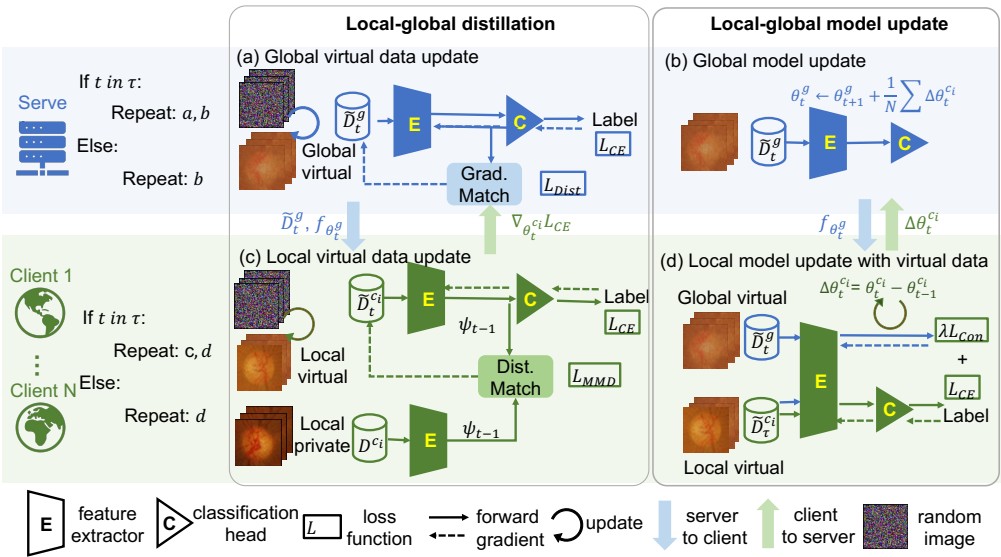

Figure 2: Overview pipeline for FEDLGD. We assume $T$ FL rounds will be performed, among which we will define the selected distillation rounds as $\tau \in [T]$ for local-global iteration. For selected rounds ($t \in \tau$), clients will update local models (d) and refine the local virtual data with the latest network parameters (c), while the server uses aggregated gradients from cross-entropy loss ($\mathcal{L}_{\text{CE}}$) to update global virtual data (a) and update the global model (b). We term this procedure Iterative Local-global Distillation. For the unselected rounds ($t \in T \backslash \tau$), we perform ordinary FL pipeline on local virtual data with regularization loss ($\mathcal{L}_{\text{Con}}$) on global virtual data.

with parameters $\theta$ on the distributed datasets ($D \equiv \bigcup_{i \in [N]} D_i$). The objective function is written as:

$$\mathcal{L}(\theta) = \sum_{i=1}^{N} \frac{|D_i|}{|D|} \mathcal{L}_i(\theta), \tag{1}$$

where $\mathcal{L}_i(w)$ is the empirical loss of client $i$.

In practice, different clients in FL may have variant amounts of training samples, leading to asynchronized updates. In this work, we focus on a new type of FL training method – federated virtual learning, that trains on distilled datasets for efficiency and synchronization (discussed in Sec. 2.3) Federated virtual learning synthesizes local virtual data $\tilde{D}_i$ for client $i$ for $i \in [N]$ and form $\tilde{D} \equiv \bigcup_{i \in [N]} \tilde{D}_i$. Typically, $|\tilde{D}_i| \ll |D_i|$ and $|\tilde{D}_i| = |\tilde{D}_j|$. A basic setup for federated virtual learning is to replace $D_i$ with $\tilde{D}_i$ in Eq (1), namely FL model is trained on the virtual datasets. As suggested in FedDM [40], the clients should not share gradients w.r.t. the original data for privacy concern.

## 3.2 Overall Pipeline

The overall pipeline of our proposed method contains three stages, including *1) initialization, 2) iterative local-global distillation, and 3) federated virtual learning.* We depict the overview of FEDLGD pipeline in Figure 2. However, FL is inevitability affected by several challenges, including synchronization, efficiency, privacy, and heterogeneity. Specifically, we outline FEDLGD as follows:

We begin with the initialization of the clients' local virtual data $\tilde{D}^c$ by performing initial rounds of distribution matching (DM) [45]. Meanwhile, the server will initialize global virtual data $\tilde{D}^g$ and network parameters $\theta_0^g$. In this stage, we generate the same amount of class-balanced virtual data for each client and server.

Then, we will refine our local and global virtual data using our proposed *local-global* distillation strategies in Sec. 3.3.1 and 3.3.2. This step is performed for a few selected iterations (e.g. $\tau = \{0, 5, 10\}$) to update $\theta$ using $\mathcal{L}_{\text{CE}}$ (Eq 3), $\tilde{D}^g$ using $\mathcal{L}_{\text{Dist}}$ (Eq 5), and $\tilde{D}^c$ using $\mathcal{L}_{\text{MMD}}$ (Eq 2) in early training epochs. For each selected iterations, the server and clients will update their virtual data for a few distillation steps.

Finally, after refining local and global virtual data $\tilde{D}^g$ and $\tilde{D}^c$, we continue federated virtual learning in stage 3 on local virtual data $\tilde{D}^c$ using $\mathcal{L}_{\text{total}}$ (Eq 3), with $\tilde{D}^g$ as regularization anchor to calculate $\mathcal{L}_{\text{Con}}$ (Eq. 4). We provide implementation details, an algorithm box, and an anonymous link to our code in the Appendix.

### 3.3 FL with Local-Global Dataset Distillation

#### 3.3.1 Local Data Distillation

Our purpose is to decrease the number of local data to achieve efficient training to meet the following goals. First of all, we hope to synthesize virtual data conditional on class labels to achieve class-balanced virtual datasets. Second, we hope to distill local data that is best suited for the classification task. Last but not least, the process should be efficient due to the limited computational resource locally. To this end, we design Iterative Distribution Matching to fulfill our purpose.

**Iterative distribution matching.** We aim to gradually improve distillation quality during FL training. To begin with, we split a model into two parts, feature extractor $\psi$ (shown as $E$ in figure 2) and classification head $h$ (shown as $C$ in figure 2). The whole classification model is defined as $f^\theta = h \circ \psi$. The high-level idea of distribution matching can be described as follows. Given a feature extractor $\psi : \mathbb{R}^d \to \mathbb{R}^{d'}$, we want to generate $\tilde{D}$ so that $P_\psi(D) \approx P_\psi(\tilde{D})$ where $P$ is the distribution in feature space. To distill local data during FL efficiently that best fits our task, we intend to use the up-to-date server model's feature extractor as our kernel function to distill better virtual data. Since we can't obtain ground truth distribution of local data, we utilize empirical maximum mean discrepancy (MMD) [11] as our loss function for local virtual distillation:

$$\mathcal{L}_{\text{MMD}} = \sum_{k}^{K} || \frac{1}{|D_k^c|} \sum_{i=1}^{|D_k^c|} \psi^t(x_i) - \frac{1}{|\tilde{D}_k^{c,t}|} \sum_{j=1}^{|\tilde{D}_k^{c,t}|} \psi^t(\tilde{x}_j^t) ||^2, \quad (2)$$

where $\psi^t$ and $\tilde{D}^{c,t}$ are the server feature extractor and local virtual data from the latest global iteration $t$. Following [46, 45], we apply the differentiable Siamese augmentation on virtual data $\tilde{D}^c$. $K$ is the total number of classes, and we sum over MMD loss calculated per class $k \in [K]$. In such a way, we can generate balanced local virtual data by optimizing the same number of virtual data per class.

Although such an efficient distillation strategy is inspired by DM [45], we highlight the key difference that DM uses randomly initialized deep neural networks to extract features, whereas we use trained FL models with task-specific supervised loss. We believe *iterative updating* on the clients' data using the up-to-date network parameters can generate better task-specific local virtual data. Our intuition comes from the recent success of the empirical neural tangent kernel for data distribution learning and matching [30, 8]. Especially, the feature extractor of the model trained with FEDLGD could obtain feature information from other clients, which further harmonize the domain shift between clients. We apply DM [45] to the baseline FL methods and demonstrate the effectiveness of our proposed iterative strategy in Sec. 4. Furthermore, note that FEDLGD only requires a few hundreds of local distillations steps using the local model's feature distribution, which is more computationally efficient than other bi-level dataset distillation methods [46, 5].

**Harmonizing local heterogeneity with global anchors.** Data collected in different sites may have different distributions due to different collecting protocols and populations. Such heterogeneity will degrade the performance of FL. Worse yet, we found increased data heterogeneity among clients when federatively training with distilled local virtual data (see Figure 1). We aim to alleviate the dataset shift by adding a regularization term in feature space to our total loss function for local model updating, which is inspired by [37, 20]:

$$\mathcal{L}_{\text{total}} = \mathcal{L}_{\text{CE}}(\tilde{D}^g, \tilde{D}^c; \theta) + \lambda \mathcal{L}_{\text{Con}}(\tilde{D}^g, \tilde{D}^c), \quad (3)$$

and

$$\mathcal{L}_{\text{Con}} = \sum_{i \in I} \frac{-1}{|P(i)|} \sum_{p \in P(i)} \log \frac{\exp(z_g \cdot z_p / \tau_{temp})}{\sum_{a \in A(i)} \exp(z_g \cdot z_a / \tau_{temp})}, \quad (4)$$

where $\mathcal{L}_{\text{CE}}$ is the cross-entropy measured on the virtual data, and $\mathcal{L}_{\text{Con}}$ is the supervised contrastive loss where $I$ is the collection of all indices, $A(i)$ indicates all the local and global virtual data indices without $i$ (i.e. $A(i) \equiv I \backslash \{i\}$), $z = \psi(x)$ is the output of feature extractor, $P(i)$ represents the set of

images belonging to the same class $y_i$ without data $i$, and $\tau_{temp}$ is a scalar temperature parameter. In such a way, global virtual data can be served for calibration, where $z_g$ is from $\tilde{D}^g$ as an anchor, and $z_p$ and $z_a$ are from $\tilde{D}^c$. At this point, a critical problem arises: *What global virtual data shall we use?*

### 3.3.2 Global Data Distillation

Here, we provide an affirmative solution to the question of generating global virtual data that can be naturally incorporated into FL pipeline. Although distribution-based matching is efficient, local clients may not share their features due to privacy concerns. Therefore, we propose to leverage local clients' averaged gradients to distill global virtual data and utilize it in Eq. (4). We term our global data distillation method as *Federated Gradient Matching*.

**Federated gradient matching.** The concept of gradient-based dataset distillation is to minimize the distance between gradients from model parameters trained by original data and distilled data. It is usually considered as a learning-to-learn problem because the procedure consists of model updates and distilled data updates. Zhao *et al.* [46] studies gradient matching in the centralized setting via bi-level optimization that iteratively optimizes the virtual data and model parameters. However, the implementation in [46] is not appropriate for our specific context because there are two fundamental differences in our settings: 1) for model updating, the gradient-distilled dataset is on the server and will not directly optimize the targeted task; 2) for virtual data update, the 'optimal' model comes from the optimized local model aggregation. These two steps can naturally be embedded in local model updating and global virtual data distillation from the aggregated local gradients. First, we utilize the distance loss $\mathcal{L}_{Dist}$ [46] for gradient matching:

$$\mathcal{L}_{Dist} = Dist(\bigtriangledown_\theta \mathcal{L}_{CE}^{\tilde{D}^g}(\theta), \bigtriangledown_\theta \mathcal{L}_{CE}^{\tilde{D}^c}(\theta)) \tag{5}$$

where $\tilde{D}^c$ and $\tilde{D}^g$ denote local and global virtual data, $\bigtriangledown_\theta \mathcal{L}_{CE}^{\tilde{D}^c}$ is the average client gradient. Then, our proposed federated gradient matching optimize as follows:

$$\min_{D^g} \mathcal{L}_{Dist}(\theta) \quad \text{subject to} \quad \theta = \frac{1}{N}\theta^{c_i*},$$

where $\theta^{c_i*} = \arg\min_\theta \mathcal{L}_i(\tilde{D}^c)$ is the optimal local model weights of client $i$ at a certain round $t$.

Noting that compared with FedAvg [29], there is no additional client information shared for global distillation. We also note the approach seems similar to the gradient inversion attack [49] but we consider averaged gradients w.r.t. local virtual data, and the method potentially defenses inference attack better (Appendix D.8), which is also implied by [40, 7]. Privacy preservation can be further improved by employing differential privacy [1], but this is not the main focus of our work.

## 4 Experiment

To evaluate FEDLGD, we consider the FL setting in which clients obtain data from different domains while performing the same task. Specifically, we compare with multiple baselines on benchmark datasets DIGITS (Sec. 4.2), where each client has data from completely different open-sourced datasets. The experiment is designed to show that FEDLGD can effectively mitigate large domain shifts. Additionally, we evaluate the performance of FEDLGD on another benchmark dataset, CIFAR10C [14], which collects data from different corrupts yielding data distribution shift and contains a large number of clients, so that we can investigate varied client sampling in FL. The experiment aims to show FEDLGD's feasibility on large-scale FL environments. We also validate the performance under medical datasets, RETINA, in Appendix. B.

### 4.1 Training and Evaluation Setup

**Model architecture.** We conduct the ablation study to explore the effect of different deep neural networks' performance under FEDLGD. Specifically, we adapt ResNet18 [13] and ConvNet [46] in our study. To achieve the optimal performance, we apply the same architecture to perform both the local distillation task and the classification task, as this combination is justified to have the best output [46, 45]. The detailed model architectures are presented in Appendix D.4.

**Comparison methods.** We compare the performance of downsteam classification tasks using state-of-the-art (SOTA) FL algorithms, FedAvg [29], FedProx [26], FedNova [38], Scaffold [18], MOON [24],

Table 1: Test accuracy for `DIGITS` under different images per class (IPC) and model architectures. R and C stand for ResNet18 and ConvNet, respectively, and we set IPC to 10 and 50. Threre are five clients (MNIST, SVHN, USPS, SynthDigits, and MNIST-M) containing data from different domains. 'Average' is the unweighted test accuracy average of all the clients. The best performance under different models is highlighted using **bold**. The best results on ConvNet are marked in red and in black for ResNet18.

| DIGITS | | MNIST | | SVHN | | USPS | | SynthDigits | | MNIST-M | | Average | |
|---|---|---|---|---|---|---|---|---|---|---|---|---|---|
| IPC | | 10 | 50 | 10 | 50 | 10 | 50 | 10 | 50 | 10 | 50 | 10 | 50 |
| FedAvg | R | 73.0 | 92.5 | 20.5 | 48.9 | 83.0 | 89.7 | 13.6 | 28.0 | 37.8 | 72.3 | 45.6 | 66.3 |
| | C | 94.0 | 96.1 | 65.9 | 71.7 | 91.0 | 92.9 | 55.5 | 69.1 | 73.2 | 83.3 | 75.9 | 82.6 |
| FedProx | R | 72.6 | 92.5 | 19.7 | 48.4 | 81.5 | 90.1 | 13.2 | 27.9 | 37.3 | 67.9 | 44.8 | 65.3 |
| | C | 93.9 | 96.1 | 66.0 | 71.5 | 90.9 | 92.9 | 55.4 | 69.0 | 73.7 | 83.3 | 76.0 | 82.5 |
| FedNova | R | 75.5 | 92.3 | 17.3 | 50.6 | 80.3 | 90.1 | 11.4 | 30.5 | 38.3 | 67.9 | 44.6 | 66.3 |
| | C | 94.2 | 96.2 | 65.5 | 73.1 | 90.6 | 93.0 | 56.2 | 69.1 | 74.6 | 83.7 | 76.2 | 83.0 |
| Scaffold | R | 75.8 | 93.4 | 16.4 | 53.8 | 79.3 | 91.3 | 11.2 | 34.2 | 38.3 | 70.8 | 44.2 | 68.7 |
| | C | 94.1 | 96.3 | 64.9 | 73.3 | 90.6 | 93.4 | 56.0 | 70.1 | 74.6 | 84.7 | 76.0 | 83.6 |
| MOON | R | 15.5 | 80.4 | 15.9 | 14.2 | 25.0 | 82.4 | 10.0 | 11.5 | 11.0 | 35.4 | 15.5 | 44.8 |
| | C | 85.0 | 95.5 | 49.2 | 70.5 | 83.4 | 92.0 | 31.5 | 67.2 | 56.9 | 82.3 | 61.2 | 81.5 |
| VHL | R | 87.8 | 95.9 | 29.5 | 67.0 | 88.0 | 93.5 | 18.2 | 60.7 | 52.2 | 85.7 | 55.1 | 80.5 |
| | C | 95.0 | 96.9 | 68.6 | 75.2 | 92.2 | 94.4 | 60.7 | 72.3 | 76.1 | 83.7 | 78.5 | 84.5 |
| FEDLGD | R | **92.9** | **96.7** | **46.9** | **73.3** | **89.1** | **93.9** | **27.9** | **72.9** | **70.8** | **85.2** | **65.5** | **84.4** |
| | C | **95.8** | **97.1** | 68.2 | **77.3** | **92.4** | **94.6** | **67.4** | **78.5** | **79.4** | **86.1** | **80.6** | **86.7** |

and VHL [37][2]. We directly use local virtual data from our initialization stage for FL methods other than ours. We perform classification on client's testing set and report the test accuracies.

**FL training setup.** We use the SGD optimizer with a learning rate of $10^{-2}$ for `DIGITS` and `CIFAR10C`. If not specified, our default setting for local model update epochs is 1, total update rounds is 100, the batch size for local training is 32, and the number of virtual data update iterations ($|\tau|$) is 10. The numbers of default virtual data distillation steps for clients and server are set to 100 and 500, respectively. Since we only have a few clients for `DIGITS` and `RETINA` experiments, we will select all the clients for each iteration, while the client selection for `CIFAR10C` experiments will be specified in Sec. 4.3. The experiments are run on NVIDIA GeForce RTX 3090 Graphics cards with PyTorch.

**Proper Initialization for Distillation.** We propose to initialize the distilled data using statistics from local data to take care of both privacy concerns and model performance. Specifically, each client calculates the statistics of its own data for each class, denoted as $\mu_i^c, \sigma_i^c$, and then initializes the distillation images per class, $x \sim \mathcal{N}(\mu_i^c, \sigma_i^c)$, where $c$ and $i$ represent each client and categorical label. The server only needs to aggregate the statistics and initializes the virtual data as $x \sim \mathcal{N}(\mu_i^g, \sigma_i^g)$. In this way, no real data is shared with any participant in the FL system. The comparison results using different initialization methods proposed in previous works [46, 45] can be found in Appendix C.

## 4.2 `DIGITS` Experiment

**Datasets.** We use the following datasets for our benchmark experiments: `DIGITS` = {MNIST [21], SVHN [31], USPS [17], SynthDigits [9], MNIST-M [9]}. Each dataset in `DIGITS` contains hand-written, real street and synthetic digit images of $0, 1, \cdots, 9$. As a result, we have 5 clients in the experiments, and image size is $28 \times 28$.

**Comparison with baselines under various conditions.** To validate the effectiveness of FEDLGD, we first compare it with the alternative FL methods varying on two important factors: Image-per-class (IPC) and different deep neural network architectures (arch). We use IPC $\in \{10, 50\}$ and arch $\in$ { ResNet18(R), ConvNet(C)} to examine the performance of SOTA models and FEDLGD using distilled `DIGITS`. Note that we fix IPC = 10 for global virtual data and vary IPC for local virtual data. Table 1 shows the test accuracies of `DIGITS` experiments. In addition to testing with original test sets, we also show the unweighted averaged test accuracy. One can observe that for each FL algorithm, ConvNet(C) always has the best performance under all IPCs. The observation is consistent with [45] as more complex architectures may cause over-fitting in training virtual data. It is also shown that using IPC = 50 always outperforms IPC = 10 as expected since more data are available for training. Overall, FEDLGD outperforms other SOTA methods, where on average accuracy, FEDLGD increases the best test accuracy results among the baseline methods of 2.1% (IPC =10, arch = C), 10.4% (IPC

---

[2]The detailed information of the methods can be found in Appendix E.

Table 2: Averaged test accuracy for `CIFAR10C` with ConvNet.

| CIFAR10C | | FedAvg | | FedProx | | FedNova | | Scaffold | | MOON | | VHL | | FEDLGD | |
|---|---|---|---|---|---|---|---|---|---|---|---|---|---|---|---|---|
| IPC | | 10 | 50 | 10 | 50 | 10 | 50 | 10 | 50 | 10 | 50 | 10 | 50 | 10 | 50 |
| | 0.2 | 27.0 | 44.9 | 27.0 | 44.9 | 26.7 | 34.1 | 27.0 | 44.9 | 20.5 | 31.3 | 21.8 | 45.0 | **32.9** | **46.8** |
| Client ratio | 0.5 | 29.8 | 51.4 | 29.8 | 51.4 | 29.6 | 45.9 | 30.6 | 51.6 | 23.8 | 43.2 | 29.3 | 51.7 | **39.5** | **52.8** |
| | 1 | 33.0 | 54.9 | 33.0 | 54.9 | 30.0 | 53.2 | 33.8 | 54.5 | 26.4 | 51.6 | 34.4 | 55.2 | **47.6** | **57.4** |

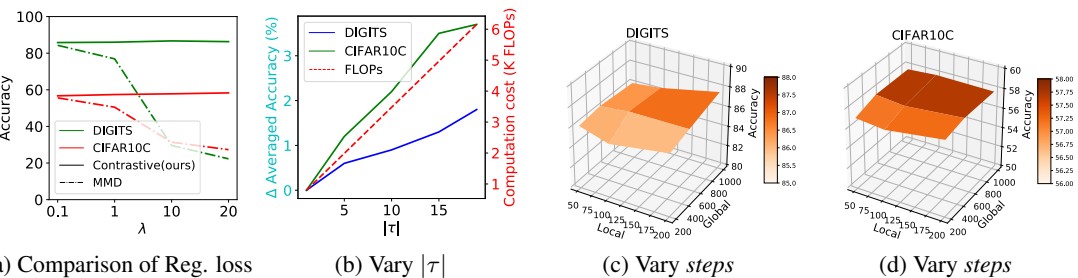

(a) Comparison of Reg. loss    (b) Vary $|\tau|$    (c) Vary *steps*    (d) Vary *steps*

Figure 3: (a) Comparison between different regularization losses and their weighting in total loss ($\lambda$). One can observe that supervised contrastive loss gives us better and more stable performance with different coefficient choices. (b) The trade-off between $|\tau|$ and computation cost. One can observe that the model performance improves with the increasing $|\tau|$, which is a trade-off between computation cost and model performance. Vary data updating *steps* for (c) `DIGITS` and (d) `CIFAR10C`. One can observe that FEDLGD yields consistent performance, and the accuracy tends to improve with an increasing number of local and global steps.

=10, arch = R), 2.2% (IPC = 50, arch = C) and 3.9% (IPC =50, arch = R). VHL [37] is the closest strategy to FEDLGD and achieves the best performance among the baseline methods, indicating that the feature alignment solutions are promising for handling heterogeneity in federated virtual learning. However, VHL is still worse than FEDLGD, and the performance may result from the differences in synthesizing global virtual data. VHL [37] uses untrained StyleGAN [19] to generate global virtual data without further updating. On the contrary, we update our global virtual data during FL training.

### 4.3 `CIFAR10C` Experiment

**Datasets.** We conduct real-world FL experiments on `CIFAR10C`[3], where, like previous studies [24], we apply Dirichlet distribution with $\alpha = 0.5$ to generate 3 partitions on each distorted Cifar10-C [14], resulting in 57 clients each with class imbalanced non-IID datasets. In addition, we apply random client selection with ratio = 0.2, 0.5, and 1 and set image size as $28 \times 28$.

**Comparison with baselines under different client sampling ratios.** The objective of the experiment is to test FEDLGD under popular FL questions: class imbalance, large number of clients, different client sample ratios, and data heterogeneity. One benefit of federated virtual learning is that we can easily handle class imbalance by distilling the same number (IPC) of virtual data. We will vary IPC and fix the model architecture to ConvNet since it is validated that ConvNet yields better performance in virtual training [46, 45]. One can observe from Table 2 that FEDLGD consistently achieves the best performance under different IPC and client sampling ratios. We would like to point out that when IPC=10, the performance boosts are significant, which indicates that FEDLGD is well-suited for FL conditions when there is a large group of clients and each of them has a limited number of data.

### 4.4 Ablation studies for FEDLGD

The success of FEDLGD relies on the novel design of local-global data distillation, where the selection of regularization loss and the number of iterations for data distillation plays a key role. In this section, we study the choice of regularization loss and its weighting ($\lambda$) in the total loss function. Recall that among the total FL training epochs, we perform local-global distillation on the selected $\tau$ *iterations*, and within each selected *iteration*, the server and clients will perform data updating

---

[3]Cifar10-C is a collection of augmented Cifar10 that applies 19 different corruptions.

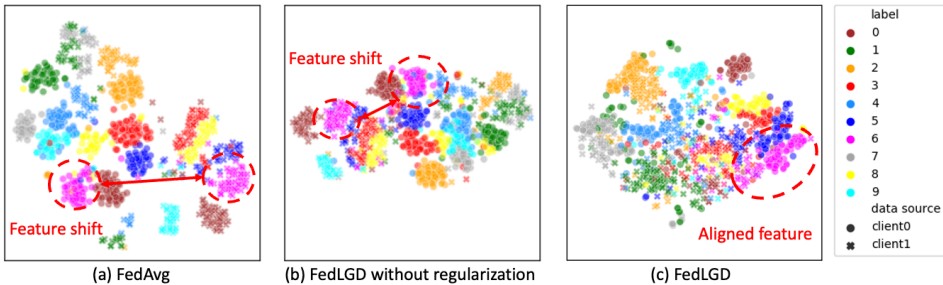

Figure 4: tSNE plots on feature space for FedAvg, FEDLGD without regularization, and FEDLGD. One can observe regularizing training with our global virtual data can rectify feature shift among different clients.

for some pre-defined *steps*. The effect of local-global distillation *iterations* and data updating *steps* will also be discussed. We also perform additional ablation studies such as computation cost and communication overhead in Appendix C.

**Effect of regularization loss.** FEDLGD uses supervised contrastive loss $\mathcal{L}_{\mathrm{Con}}$ as a regularization term to encourage local and global virtual data embedding into a similar feature space. To demonstrate the effectiveness of the regularization term in FEDLGD, we perform ablation studies to replace $\mathcal{L}_{\mathrm{Con}}$ with an alternative distribution similarity measurement, MMD loss, with different $\lambda$'s ranging from 0.1 to 20. Figure 3a shows the average test accuracy. Using supervised contrastive loss gives us better and more stable performance with different coefficient choices.

To explain the effect of our proposed regularization loss on feature representations, we embed the latent features before fully-connected layers to a 2D space using tSNE [28] shown in Figure 4. For the model trained with FedAvg (Figure 4(a)), features from two clients ($\times$ and $\circ$) are closer to their own distribution regardless of the labels (colors). In Figure 4(b), we perform virtual FL training but without the regularization term (Eq. 4). Figure 4(c) shows FEDLGD, and one can observe that data from different clients with the same label are grouped together. This indicates that our regularization with global virtual data is useful for learning homogeneous feature representations.

**Analysis of distillation *iterations* ($|\tau|$).** Figure 3b shows the improved averaged test accuracy if we increase the number of distillation iterations with FEDLGD. The base accuracy for DIGITS and CIFAR10C are 85.8 and 55.2, respectively. We fix local and global update *steps* to 100 and 500, and the selected iterations ($\tau$) are defined as arithmetic sequences with $d = 5$ (i.e., $\tau = \{0, 5, ...\}$). One can observe that the model performance improves with the increasing $|\tau|$. This is because we obtain better virtual data with more local-global distillation iterations, which is a trade-off between computation cost and model performance. We select $|\tau| = 10$ for efficiency trade-off.

**Robustness on virtual data update *steps*.** In Figure 3c and Figure 3d, we fix $|\tau| = 10$, and vary (local, global) data updating steps. One can observe that FEDLGD yields stable performance, and the accuracy slightly improves with an increasing number of local and global steps. Nevertheless, the results are all the best when comparing with the baselines. It is also worth noting that there is still trade-off between *steps* and computation cost (See Appendix).

## 5   Conclusion

In this paper, we introduce a new approach for FL, called FEDLGD. It utilizes virtual data on both client and server sides to train FL models. We are the first to reveal that FL on local virtual data can increase heterogeneity. Furthermore, we propose iterative distribution matching and federated gradient matching to iteratively update local and global virtual data, and apply global virtual regularization to effectively harmonize domain shift. Our experiments on benchmark and real medical datasets show that FEDLGD outperforms current state-of-the-art methods in heterogeneous settings. Furthermore, FEDLGD can be combined with other heterogenous FL methods such as FedProx [26] and Scaffold [18] to further improve its performance. The potential limitation lies in the additional communication and computation cost in data distillation, but we show that the trade-off is acceptable and can be mitigated by decreasing distillation *iterations* and *steps*. Our future direction will be investigating privacy-preserving data generation. We believe that this work sheds light on how to effectively mitigate data heterogeneity from a dataset distillation perspective and will inspire future work to enhance FL performance, privacy, and efficiency.

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
