**Road Map of Appendix** Our appendix is organized into five sections. The notation table is in Appendix A, which contains the mathematical notation and Algorithm 1, which outlines the pipeline of FEDLGD. Appendix B shows the results for RETINA, a real-world medical dataset. Appendix C provides a list of ablation studies to analyze FEDLGD, including computation cost, communication overhead, convergence rate, and hyper-parameter choices. Appendix D lists the details of our experiments: D.1 visualizes the original sample images used in our experiments; D.2 visualizes the local and global distilled images; D.3 shows the pixel histogram for the DIGITS and RETINA datasets for visualizing the heterogeneity of them; D.4 shows the model architectures that we used in the experiments; D.5 contains the hyper-parameters that we used to conduct all experiments; D.6 provides experiments and analysis for the privacy of FEDLGD through membership inference attack. Finally, Appendix E provides a detailed literature review and implementation of the state-of-the-art heterogeneous FL strategies. Our code and model checkpoints are available in this anonymous link: https://drive.google.com/drive/folders/1Hpy8kgPtxC_NMqK6eALwukFZJB7yf8Vl?usp=sharing[4].

# A   Notation Table

Table 3: Important notations used in the paper.

| Notations | Description |
|---|---|
| $d$ | input dimension |
| $d'$ | feature dimension |
| $f^\theta$ | global model |
| $\theta$ | model parameters |
| $\psi$ | feature extractor |
| $h$ | projection head |
| $D^g, D^c$ | original global and local data |
| $\tilde{D}^g, \tilde{D}^c$ | global and local synthetic data |
| $\tilde{f}^g, \tilde{f}^c$ | features of global and local synthetic data |
| $\mathcal{L}_{\text{total}}$ | total loss function for virtual federated training |
| $\mathcal{L}_{\text{CE}}$ | cross-entropy loss |
| $\mathcal{L}_{\text{Dist}}$ | Distance loss for gradient matching |
| $\mathcal{L}_{\text{MMD}}$ | MMD loss for distribution matching |
| $\mathcal{L}_{\text{Con}}$ | Contrastive loss for local training regularization |
| $\lambda$ | coefficient for local training regularization term |
| $T$ | total training iterations |
| $T_{\text{D}}^{\text{c}}$ | local data updating iterations for each call |
| $T_{\text{D}}^{\text{g}}$ | global data updating iterations for each call |
| $\tau$ | local global distillation iterations |

---

[4]The link was created by a new and anonymous account without leaking any identifiable information.

**Algorithm 1** Federated Virtual Learning with Local-global Distillation

---

**Require:** $f^\theta$: Model, $\psi^\theta$: Feature extractor, $\theta$: Model parameters, $\tilde{D}$: Virtual data, $D$: Original data, $\mathcal{L}$: Losses, $G$: Gradients.

**Distillation Functions:**
$\tilde{D}^{\mathrm{c}} \leftarrow \mathrm{DistributionMatch}(D^{\mathrm{c}}, f^\theta)$
$\tilde{D}^{\mathrm{c}}_{\mathrm{t}} \leftarrow \mathrm{IterativeDistributionMatch}(\tilde{D}^{\mathrm{c}}_{\mathrm{t}-1}, f^\theta_{\mathrm{t}})$
$\tilde{D}^{\mathrm{g}}_{\mathrm{t}+1} \leftarrow \mathrm{FederatedGradientMatch}(\tilde{D}^{\mathrm{g}}_{\mathrm{t}}, G^{\mathrm{g}}_{\mathrm{t}})$

**Initialization:**
$\tilde{D}^{\mathrm{c}}_{0} \leftarrow \mathrm{DistributionMatch}(D^{\mathrm{c}}_{\mathrm{rand}}, f^\theta_{\mathrm{rand}})$           ▷ Distilled local data for virtual FL training

**FEDLGD Pipeline:**
**for** $t = 1, \ldots, T$ **do**
    **Clients:**
    **for** each selected Client **do**
        **if** $t \in \tau$ **then**                               ▷ Local-global distillation
            $\tilde{D}^{\mathrm{c}}_{\mathrm{t}} \leftarrow \mathrm{IterativeDistributionMatch}(\tilde{D}^{\mathrm{c}}_{\mathrm{t}-1}, f^\theta_{\mathrm{t}})$
            $G^{\mathrm{c}}_{\mathrm{t}} \leftarrow \nabla_\theta \mathcal{L}_{\mathrm{CE}}(\tilde{D}^{\mathrm{c}}_{\mathrm{t}}, f^\theta_{\mathrm{t}})$
        **else**
            $\tilde{D}^{\mathrm{c}}_{\mathrm{t}} \leftarrow \tilde{D}^{\mathrm{c}}_{\mathrm{t}-1}$
            $G^{\mathrm{c}}_{\mathrm{t}} \leftarrow \nabla_\theta \left( \mathcal{L}_{\mathrm{CE}}(\tilde{D}^{\mathrm{c}}_{\mathrm{t}}, f^\theta_{\mathrm{t}}) + \lambda \mathcal{L}_{\mathrm{CON}}(\psi^\theta_{\mathrm{t}}(\tilde{D}^{\mathrm{g}}_{\mathrm{t}}), \psi^\theta_{\mathrm{t}}(\tilde{D}^{\mathrm{c}}_{\mathrm{t}})) \right)$
        **end if**
        Uploads $G^{\mathrm{c}}_{t}$ to Server
    **end for**
    **Server:**
    $G^{\mathrm{g}}_{\mathrm{t}} \leftarrow \mathrm{Aggregate}(G^{1}_{\mathrm{t}}, ..., G^{\mathrm{c}}_{\mathrm{t}})$
    **if** $t \in \tau$ **then**                                   ▷ Local-global distillation
        $\tilde{D}^{\mathrm{g}}_{\mathrm{t}+1} \leftarrow \mathrm{FederatedGradientMatch}(\tilde{D}^{\mathrm{g}}_{\mathrm{t}}, G^{\mathrm{g}}_{\mathrm{t}})$
        Send $\tilde{D}^{\mathrm{g}}_{\mathrm{t}+1}$ to Clients
    **end if**
    $f^\theta_{\mathrm{t}+1} \leftarrow \mathrm{ModelUpdate}(G^{\mathrm{g}}_{\mathrm{t}}, f^\theta_{\mathrm{t}})$
    Send $f^\theta_{\mathrm{t}+1}$ to Clients
**end for**

---

## B   Experiment Results on Real-world Dataset

Table 4: Test accuracy for `RETINA` experiments under different model architectures and IPC=10. R and C stand for ResNet18 and ConvNet, respectively. We have 4 clients: Drishti(D), Acrima(A), Rim(Ri), and Refuge(Re), respectively. We also show the average test accuracy (Avg). The best results on ConvNet are marked in red and in **bold** for ResNet18. The same accuracy for different methods is due to the limited number of testing samples.

| RETINA | | D | A | Ri | Re | Avg |
|---|---|---|---|---|---|---|
| FedAvg | R | 31.6 | 71.0 | 52.0 | **78.5** | 58.3 |
| | C | 69.4 | 84.0 | 88.0 | 86.5 | 82.0 |
| FedProx | R | 31.6 | 70.0 | 52.0 | **78.5** | 58.0 |
| | C | 68.4 | 84.0 | 88.0 | 86.5 | 81.7 |
| FedNova | R | 31.6 | 71.0 | 52.0 | **78.5** | 58.3 |
| | C | 68.4 | 84.0 | 88.0 | 86.5 | 81.7 |
| Scaffold | R | 31.6 | 73.0 | 49.0 | **78.5** | 58.0 |
| | C | 68.4 | 84.0 | 88.0 | 86.5 | 81.7 |
| MOON | R | 42.1 | 71.0 | 57.0 | 70.0 | 60.0 |
| | C | 57.9 | 72.0 | 76.0 | 85.0 | 72.7 |
| VHL | R | 47.4 | 62.0 | 50.0 | 76.5 | 59.0 |
| | C | 68.4 | 78.0 | 81.0 | 87.0 | 78.6 |
| FEDLGD | R | **57.9** | **75.0** | **59.0** | 77.0 | **67.2** |
| | C | 78.9 | 86.0 | 88.0 | 87.5 | 85.1 |

**Dataset.** For medical dataset, we use the retina image datasets, `RETINA` = {Drishti [36], Acrima[6], Rim [2], Refuge [32]}, where each dataset contains retina images from different stations with image size $96 \times 96$, thus forming four clients in FL. We perform binary classification to identify *Glaucomatous* and *Normal*. Example images and distributions can be found in Appendix D.3. Each client has a held-out testing set. In the following experiments, we will use the distilled local virtual training sets for training and test the models on the original testing sets. The sample population statistics for both experiments are available in Table 12 and Table 14 in Appendix D.5.

**Comparison with baselines.** The results for `RETINA` experiments are shown in Table 4, where D, A, Ri, Re represent Drishti, Acrima, Rim, and Refuge datasets. We only set IPC=10 for this experiment as clients in `RETINA` contain much fewer data points. The learning rate is set to 0.001. The same as in the previous experiment, we vary arch $\in$ { ConvNet, ResNet18}. Similarly, ConvNet shows the best performance among architectures, and FEDLGD has the best performance compared to the other methods w.r.t the unweighted averaged accuracy (Avg) among clients. To be precise, FEDLGD increases unweighted averaged test accuracy for 3.1%(versus the best baseline) on ConvNet and 7.2%(versus the best baseline) on ResNet18, respectively. The same accuracy for different methods is due to the limited number of testing samples. We conjecture the reason why VHL [37] has lower performance improvement in `RETINA` experiments is that this dataset is in higher dimensional and clinical diagnosis evidence on fine-grained details, *e.g.*, cup-to-disc ratio and disc rim integrity [34]. Therefore, it is difficult for untrained StyleGAN [19] to serve as anchor for this kind of larger images.

# C Additional Results and Ablation Studies for FEDLGD

## C.1 Different random seeds

To show the consistent performance of FEDLGD, we repeat the experiments for `DIGITS`, `CIFAR10C`, and `RETINA` with three random seeds, and report the validation loss and accuracy curves in Figure 5 and 6 (The standard deviations of the curves are plotted as shadows.). We use ConvNet for all the experiments. IPC is set to 50 for `CIFAR10C` and `DIGITS`; 10 for `RETINA`. We use the default hyperparameters for each dataset, and only report FedAvg, FedProx, Scaffold, VHL, which achieves the best performance among baseline as indicated in Table 1, 2, and 4 for clear visualization. One can observe that FEDLGD has faster convergence rate and results in optimal performances compared to other baseline methods.

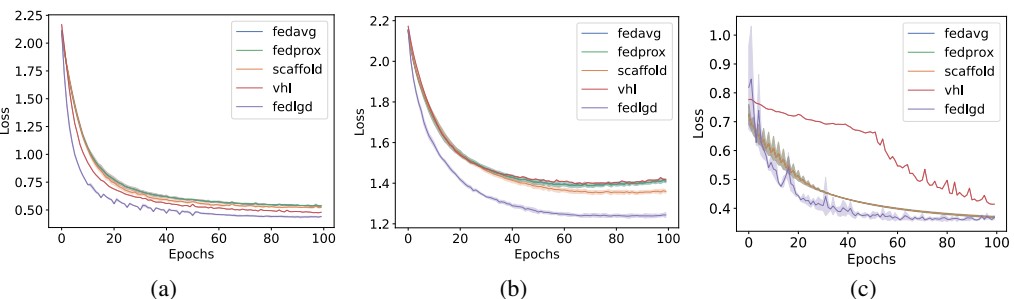

Figure 5: Averaged testing loss for (a) `DIGITS` with IPC = 50, (b) `CIFAR10C` with IPC = 50, and (c) `RETINA` with IPC = 10 experiments.

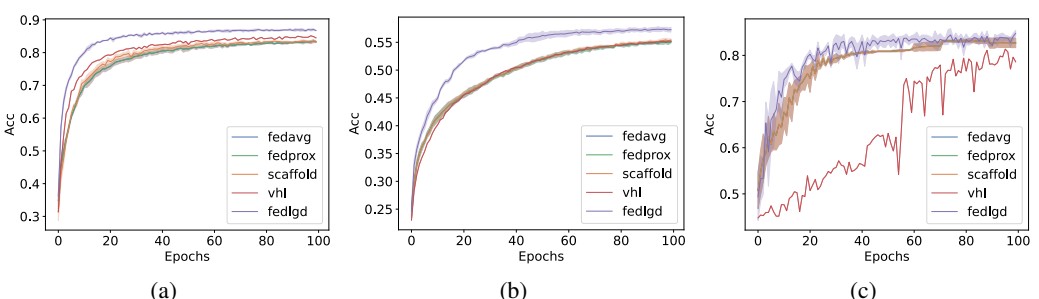

Figure 6: Averaged testing accuracy for (a) `DIGITS` with IPC = 50, (b) `CIFAR10C` with IPC = 50, and (c) `RETINA` with IPC = 10 experiments.

## C.2 Different heterogeniety levels of label shift

In the experiment presented in Sec 4.3, we study FEDLGD under both label and domain shifts, where labels are sampled from Dirichlet distribution. To ensure dataset distillation performance, we ensure that each class at least has 100 samples per client, thus setting the coefficient of Dirichlet distribution $\alpha = 2$ to simulate the worst case of label heterogeneity that meets the quality dataset distillation requirement [5]. Here, we show the performance with a less heterogeneity level ($\alpha = 5$) while keeping the other settings the same as those in Sec.4.3. The results are shown in Table 5. As we expect, the performance drop when the heterogeneity level increases ($\alpha$ decreases). One can observe that when heterogeneity increases, FEDLGD's performance drop less except for VHL. We conjecture that VHL yields similar test accuracy for $\alpha = 2$ and $\alpha = 5$ is that it uses fixed global virtual data so that the effectiveness of regularization loss does not improve much even if the heterogeneity level is decreased. Nevertheless, FEDLGD consistently outperforms all the baseline methods.

---

[5]The $\alpha$ should be 2 instead of 0.5 in Sec 4.3.

Table 5: Comparison of different $\alpha$ for Drichilet distribution on `CIFAR10C`.

| $\alpha$ | FedAvg [29] | FedProx [26] | FedNova [38] | Scaffold [18] | MOON [24] | VHL [37] | FEDLGD |
|---|---|---|---|---|---|---|---|
| 2 | 54.9 | 54.9 | 53.2 | 54.5 | 51.6 | 55.2 | **57.4** |
| 5 | 55.4 | 55.4 | 55.4 | 55.6 | 51.1 | 55.4 | **58.1** |

Table 6: Computation cost for each epoch. Nc and Ns stand for the number of updating iteration for local and global virtual data, and we defaultly set as 100 and 500, respectively. Note that we only set $|\tau| = 10$ iterations, which is a relatively small number compare to total epochs(100).

| Dataset | Vanilla FedAvg | FEDLGD(iters $\in \tau$) | FEDLGD(iters $\notin \tau$) | FEDLGD(server) |
|---|---|---|---|---|
| `DIGITS` | 238K | 2.7K + 3.4K × Nc | 4.8K | 2.9K × Ns |
| `CIFAR10C` | 53M | 2.7K + 3.4K × Nc | 4.8K | 2.9K × Ns |
| `RETINA` | 1.76M | 0.7K + 0.9K × Nc | 1K | 0.9K × Ns |

## C.3  Computation Cost

Computation cost for `DIGITS` experiment on each epoch can be found in Table 7. Nc and Ns stand for the number of updating iterations for local and global virtual data, and as default, we it set as 100 and 500, respectively. The computation costs for FEDLGD in `DIGITS` and `CIFAR10C` are identical since we used virtual data with fixed size and number for training. Plugging in the number, clients only need to operate 3.9M FLOPs for total 100 training epochs with $\tau = 10$ (our default setting), which is significantly smaller than vanilla FedAvg using original data (23.8M and 5,300M for `DIGITS` and `CIFAR10C`, respectively.).

Table 7: Communication overhead for each epoch. Note that the IPC for our global virtual data is 10, and the clients only need to *download* it for $|\tau| = 10$ times.

| Image size | ConvNet | ResNet18 | Global virtual data |
|---|---|---|---|
| 28 × 28 | 311K | 11M | 23K × IPC |
| 96 × 96 | 336K | 13M | 55K × IPC |

## C.4  Communication Overhead

The communication overhead for each epoch in `DIGITS` and `CIFAR10C` experiments are identical since we use same architectures and size of global virtual data (Table. 7 28 × 28). The analysis of `RETINA` is shown in row 96 × 96. Note that the IPC for our global virtual data is 10, and the clients only need to *download* it for $|\tau|$ times. Although FEDLGD requires clients to download additional data which is almost double the original Bytes (311K + 230K), we would like to point out that this only happens $|\tau| = 10$ times, which is a relatively small number compared to total FL training iterations.

## C.5  Analysis of batch size

Batch size is another factor for training the FL model and our distilled data. We vary the batch size $\in \{8, 16, 32, 64\}$ to train models for `CIFAR10C` with the fixed default learning rate. We show the effect of batch size in Table 8 reported on average testing accuracy. One can observe that the performance is slightly better with moderately smaller batch size which might due to two reasons: 1) more frequent model update locally; and 2) larger model update provides larger gradients, and FEDLGD can benefit from the large gradients to distill higher quality virtual data. Overall, the results are generally stable with different batch size choices.

## C.6  Analysis of Local Epoch

Aggregating at different frequencies is known as an important factor that affects FL behavior. Here, we vary the local epoch $\in \{1, 2, 5\}$ to train all baseline models on `CIFAR10C`. Figure 7 shows the result of test accuracy under different epochs. One can observe that as the local epoch increases, the performance of FEDLGD would drop a little bit. This is because doing gradient matching requires the model to be trained to an intermediate level, and if local epochs increase, the loss of `DIGITS` models

Table 8: Varying batch size in FEDLGD on CIFAR10C. We report the unweighted accuracy. One can observe that the performance increases when the batch size decreases.

| Batch Size | 8 | 16 | 32 | 64 |
|---|---|---|---|---|
| CIFAR10C | 59.5 | 58.3 | 57.4 | 56.0 |

will drop significantly. However, FEDLGD still consistently outperforms the baseline methods. As our future work, we will investigate the tuning of the learning rate in the early training stage to alleviate the effect.

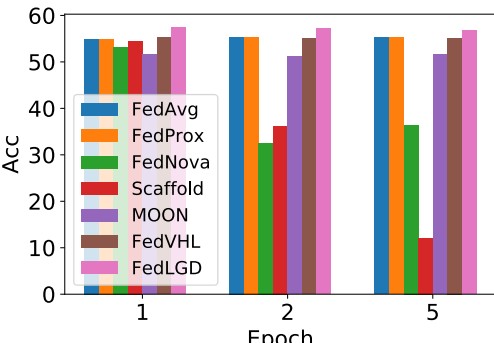

Figure 7: Comparison of model performances under different local epochs with CIFAR10C.

## C.7 Different Initialization for Virtual Images

To validate our proposed initialization for virtual images has the best trade-off between privacy and efficacy, we compare our test accuracy with the models trained with synthetic images initialized by random noise and real images in Table 9. To show the effect of initialization under large domain shift, we run experiments on DIGITS dataset. One can observe that our method which utilizes the statistics $(\mu_i, \sigma_i)$ of local clients as initialization outperforms random noise initialization. Although our performance is slightly worse than the initialization that uses real images from clients, we do not ask the clients to share real images to the server which is more privacy-preserving.

Table 9: Comparison of different initialization for synthetic images DIGITS

| CIFAR10C | MNIST | SVHN | USPS | SynthDigits | MNIST-M | Average |
|---|---|---|---|---|---|---|
| Noise ($\mathcal{N}(0,1)$) | 96.3 | 75.9 | 93.3 | 72.0 | 83.7 | 84.2 |
| Ours ($\mathcal{N}(\mu_i, \sigma_i)$) | 97.1 | 77.3 | 94.6 | 78.5 | 86.1 | 86.7 |
| Real images | 97.7 | 78.8 | 94.2 | 82.4 | 89.5 | 88.5 |

# D Experimental details

## D.1 Visualization of the original images

### D.1.1 Digits dataset

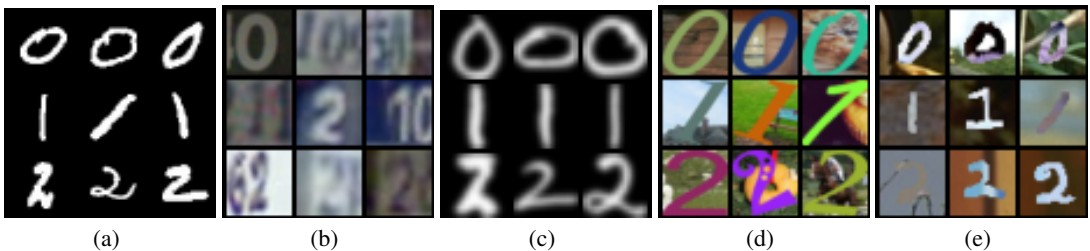

| (a) | (b) | (c) | (d) | (e) |

Figure 8: Visualization of the original digits dataset. (a) visualized the MNIST client; (b) visualized the SVHN client; (c) visualized the USPS client; (d) visualized the SynthDigits client; (e) visualized the MNIST-M client.

### D.1.2 Retina dataset

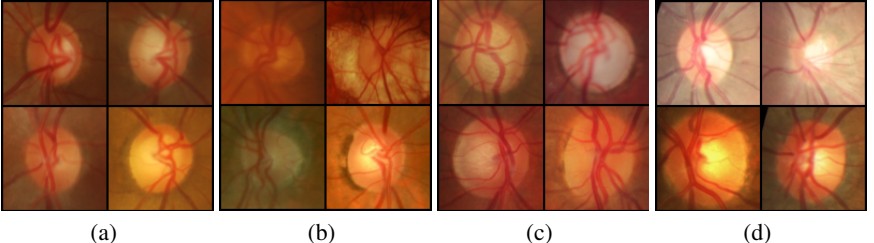

| (a) | (b) | (c) | (d) |

Figure 9: Visualization of the original retina dataset. (a) visualized the Drishti client; (b) visualized the Acrima client; (c) visualized the Rim client; (d) visualized the Refuge client.

### D.1.3 Cifar10C dataset

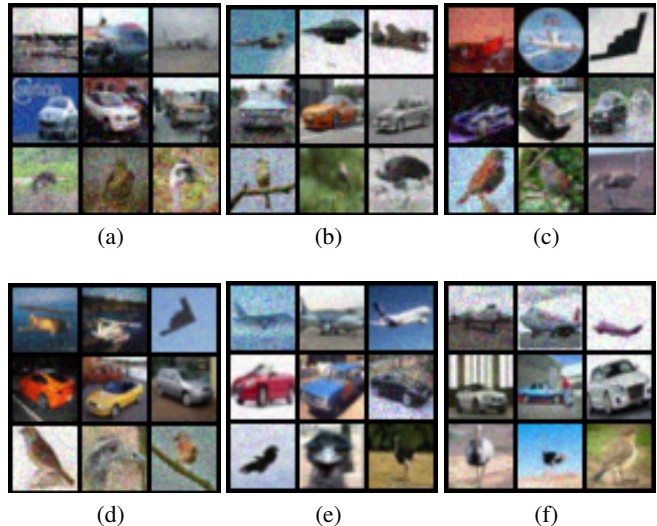

Figure 10: Visualization of the original CIFAR10C. Sampled images from the first six clients.

 ## D.2 Visualization of our distilled global and local images

 ### D.2.1 Digits dataset

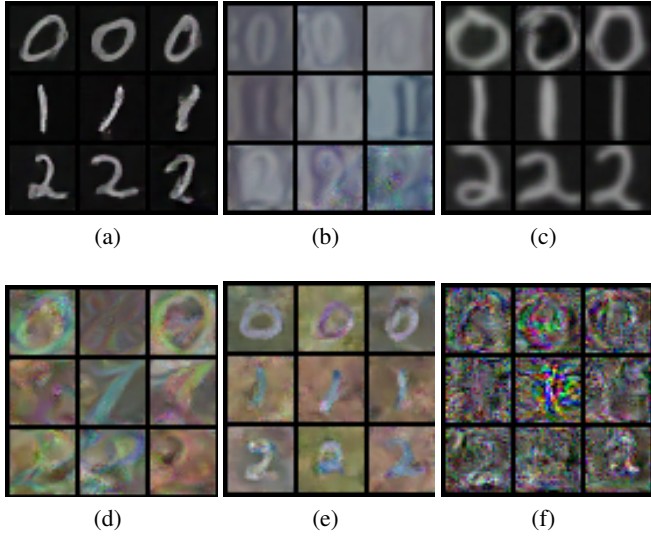

Figure 11: Visualization of the global and local distilled images from the digits dataset. (a) visualized the MNIST client; (b) visualized the SVHN client; (c) visualized the USPS client; (d) visualized the SynthDigits client; (e) visualized the MNIST-M client; (f) visualized the server distilled data.

 ### D.2.2 Retina dataset

 ### D.2.3 Cifar10C dataset

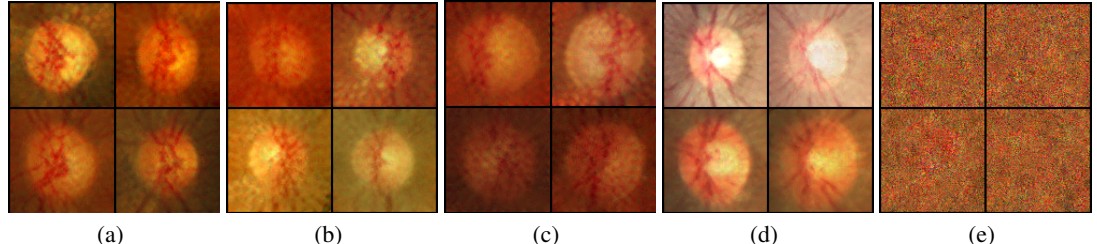

Figure 12: Visualization of the global and local distilled images from retina dataset. (a) visualized the Drishti client; (b) visualized the Acrima client; (c) visualized the Rim client; (d) visualized the Refuge client; (e) visualized the server distilled data.

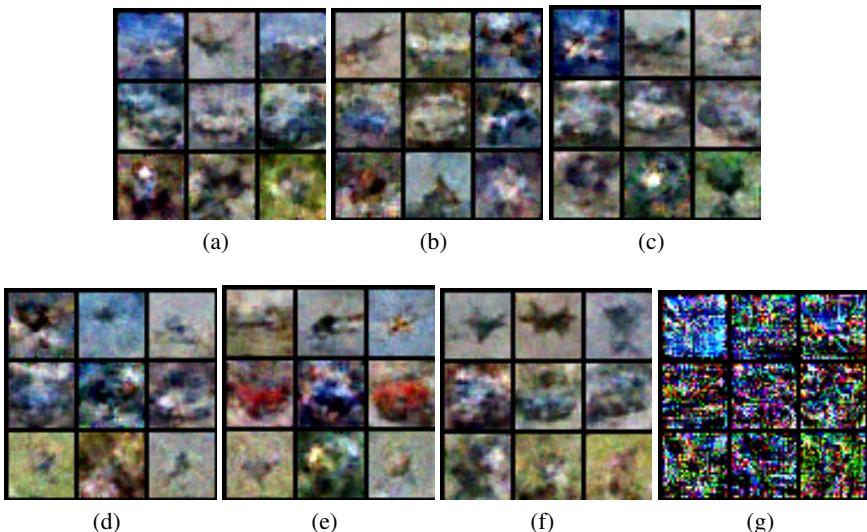

Figure 13: (a)-(f) visualizes the distailled images for the first six clients of `CIFAR10C`. (g) visualizes the global distilled images.

## D.3 Visualization of the heterogeneity of the datasets

### D.3.1 Digits dataset

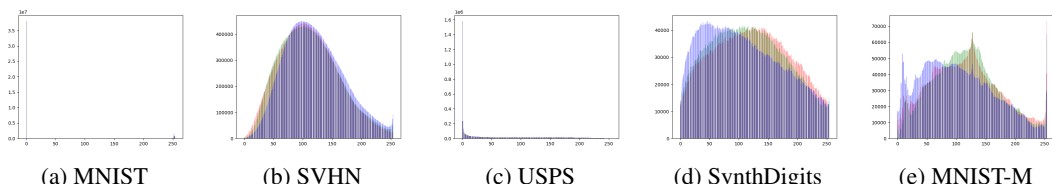

Figure 14: Histogram for the frequency of each RGB value in original `DIGITS`. The red bar represents the count for R; the green bar represents the frequency of each pixel for G; the blue bar represents the frequency of each pixel for B. One can observe the distributions are very different. Note that figure (a) and figure (c) are both greyscale images with most pixels lying in 0 and 255.

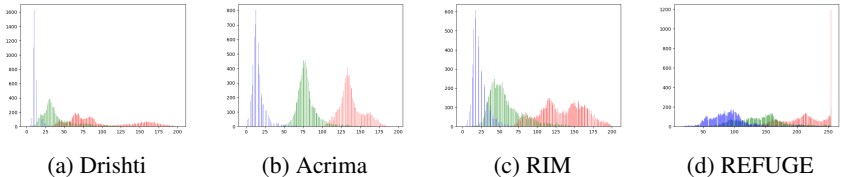

|(a) Drishti|(b) Acrima|(c) RIM|(d) REFUGE|

Figure 15: Histogram for the frequency of each RGB value in original `RETINA`. The red bar represents the count for R; the green bar represents the frequency of each pixel for G; the blue bar represents the frequency of each pixel for B.

### D.3.2 Retina dataset

### D.3.3 CIFAR10C dataset

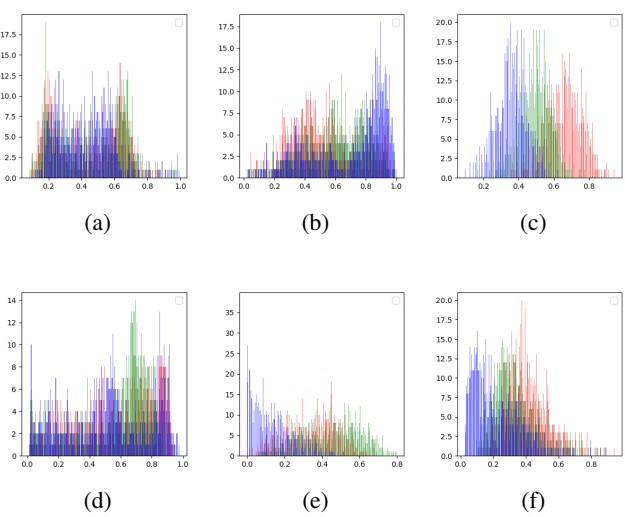

|(a)|(b)|(c)|
|(d)|(e)|(f)|

Figure 16: Histogram for the frequency of each RGB value in the first six clients of original `CIFAR10C`. The red bar represents the count for R; the green bar represents the frequency of each pixel for G; the blue bar represents the frequency of each pixel for B.

### D.4 Model architecture

For our benchmark experiments, we use ConvNet to both distill the images and train the classifier.

Table 10: ResNet 18 architecture. For the convolutional layer (Conv2D), we list parameters with a sequence of input and output dimensions, kernel size, stride, and padding. For the max pooling layer (MaxPool2D), we list kernel and stride. For a fully connected layer (FC), we list input and output dimensions. For the BatchNormalization layer (BN), we list the channel dimension.

| Layer | Details |
|-------|---------|
| 1 | Conv2D(3, 64, 7, 2, 3), BN(64), ReLU |
| 2 | Conv2D(64, 64, 3, 1, 1), BN(64), ReLU |
| 3 | Conv2D(64, 64, 3, 1, 1), BN(64) |
| 4 | Conv2D(64, 64, 3, 1, 1), BN(64), ReLU |
| 5 | Conv2D(64, 64, 3, 1, 1), BN(64) |
| 6 | Conv2D(64, 128, 3, 2, 1), BN(128), ReLU |
| 7 | Conv2D(128, 128, 3, 1, 1), BN(64) |
| 8 | Conv2D(64, 128, 1, 2, 0), BN(128) |
| 9 | Conv2D(128, 128, 3, 1, 1), BN(128), ReLU |
| 10 | Conv2D(128, 128, 3, 1, 1), BN(64) |
| 11 | Conv2D(128, 256, 3, 2, 1), BN(128), ReLU |
| 12 | Conv2D(256, 256, 3, 1, 1), BN(64) |
| 13 | Conv2D(128, 256, 1, 2, 0), BN(128) |
| 14 | Conv2D(256, 256, 3, 1, 1), BN(128), ReLU |
| 15 | Conv2D(256, 256, 3, 1, 1), BN(64) |
| 16 | Conv2D(256, 512, 3, 2, 1), BN(512), ReLU |
| 17 | Conv2D(512, 512, 3, 1, 1), BN(512) |
| 18 | Conv2D(256, 512, 1, 2, 0), BN(512) |
| 19 | Conv2D(512, 512, 3, 1, 1), BN(512), ReLU |
| 20 | Conv2D(512, 512, 3, 1, 1), BN(512) |
| 21 | AvgPool2D |
| 22 | FC(512, num_class) |

Table 11: ConvNet architecture. For the convolutional layer (Conv2D), we list parameters with a sequence of input and output dimensions, kernel size, stride, and padding. For the max pooling layer (MaxPool2D), we list kernel and stride. For a fully connected layer (FC), we list the input and output dimensions. For the GroupNormalization layer (GN), we list the channel dimension.

| Layer | Details |
|-------|---------|
| 1 | Conv2D(3, 128, 3, 1, 1), GN(128), ReLU, AvgPool2d(2,2,0) |
| 2 | Conv2D(128, 118, 3, 1, 1), GN(128), ReLU, AvgPool2d(2,2,0) |
| 3 | Conv2D(128, 128, 3, 1, 1), GN(128), ReLU, AvgPool2d(2,2,0) |
| 4 | FC(1152, num_class) |

## D.5 Training details

We provide detailed settings for experiments conducted in Table 12 for DIGITS, Table 13 for CIFAR10C, and Table 14 for RETINA.

Table 12: `DIGITS` settings for all federated learning, including the number of training and testing examples, and local update epochs. Image per class is the number of distilled images used for distribution matching only in FEDLGD.

| DataSets | MNIST | SVHN | USPS | SynthDigits | MNIST-M |
|---|---|---|---|---|---|
| Number of clients | 1 | 1 | 1 | 1 | 1 |
| Number of Training Samples | 60000 | 73257 | 7291 | 10000 | 10331 |
| Number of Testing Samples | 10000 | 26032 | 2007 | 2000 | 209 |
| Image per Class | 10,**50** | 10,**50** | 10,**50** | 10,**50** | 10,**50** |
| Local Update Epochs | **1**,2,5 | **1**,2,5 | **1**,2,5 | **1**,2,5 | **1**,2,5 |
| Local Distillation Update Epochs | 50, **100**, 200 | 50, **100**, 200 | 50, **100**, 200 | 50, **100**, 200 | 50, **100**, 200 |
| global Distillation Update Epochs | 200, **500**, 1000 | 200, **500**, 1000 | 200, **500**, 1000 | 200, **500**, 1000 | 200, **500**, 1000 |
| $\lambda$ | 10 | 10 | 10 | 10 | 10 |

Table 13: `CIFAR10C` settings for all federated learning, including the client ratio for training and testing examples, and local update epochs. Image per class is the number of distilled images used for distribution matching only in FEDLGD.

| $\alpha$ | 2 | 5 |
|---|---|---|
| Number of clients | 57 | 57 |
| Averaged Number of Training Samples | 21790 | 15000 |
| Standard Deviation of of Training Samples | 6753 | 1453 |
| Averaged Number of Testing Samples | 2419 | 1666 |
| Standard Deviation of Number of Testing Samples | 742 | 165 |
| Image per Class | 10,**50** | 10,**50** |
| Local Update Epochs | **1**,2,5 | **1**,2,5 |
| Local Distillation Update Epochs | 50, **100**, 200 | 50, **100**, 200 |
| global Distillation Update Epochs | 200, **500**, 1000 | 200, **500**, 1000 |
| $\lambda$ | 1 | 1 |

Table 14: `RETINA` settings for all federated learning, including the number of training and testing examples and local update epochs. Image per class is the number of distilled images used for distribution matching only in FEDLGD.

| Datasets | Drishti | Acrima | RIM | Refuge |
|---|---|---|---|---|
| Number of clients | 1 | 1 | 1 | 1 |
| Number of Training Samples | 82 | 605 | 385 | 1000 |
| Number of Testing Samples | 19 | 100 | 100 | 200 |
| Image per class | 10 | 10 | 10 | 10 |
| Local Distillation Update Epochs | 100 | 100 | 100 | 100 |
| global Distillation Update Epochs | 500 | 500 | 500 | 500 |
| $\lambda$ | 0.1 | 0.1 | 0.1 | 0.1 |

### D.6 Membership Inference Attack

Studies show that neural networks are prone to suffer from several privacy attacks such as Membership Inference Attacks (MIA) [35]. In MIA, the attackers have a list of *query* data, and the purpose is to determine whether the *query* data belongs to the original training set. As discussed in [7, 40], using distilled data to train a target model can defend against multiple attacks up to a certain level. We will especially apply MIA to test whether our work can defend against privacy attacks. In detail, we perform MIA directly on models trained with FedAvg (using the original data set) and FEDLGD (using the synthetic dataset). We show the attack results in Figure 17 following the evaluation in [3]. If the ROC curve intersects with the diagonal dashed line (representing a random membership classifier) or lies below it (indicating that membership inference performs worse than random chance), it signifies that the approach provides a stronger defense against membership inference compared to the method with a larger area under the ROC curve. It can be observed that models trained with synthetic data exhibit ROC curves that are more closely aligned with or positioned below the diagonal line, suggesting that attacking membership becomes more challenging.

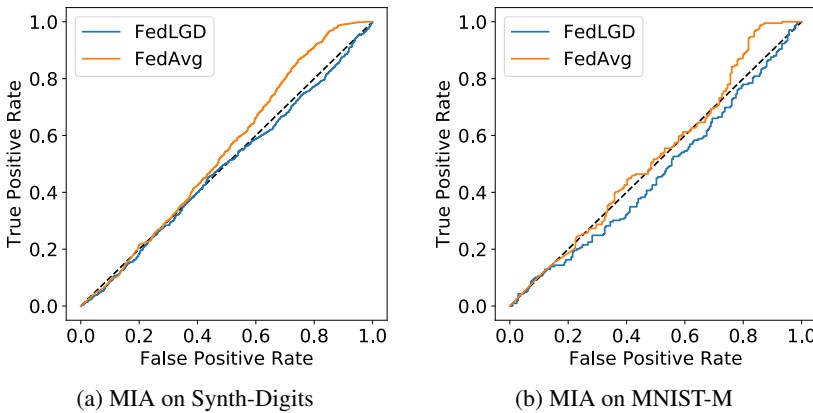

(a) MIA on Synth-Digits        (b) MIA on MNIST-M

Figure 17: MIA attack results on models trained with FedAvg (using original dataset) and FEDLGD (using distilled virtual dataset). If the ROC curve is the same as the diagonal line, it means the membership cannot be inferred. One can observe the ROC curve for the model trained with synthetic data is closer to the diagonal line, which indicates the membership information is harder to be inferred.

## E  Other Heterogeneous Federated Learning Methods Used in Comparison

FL trains the central model over a variety of distributed clients that contain non-iid data. We detailed each of the baseline methods we compared in Section 4 below.

**FedAvg [29]** The most popular aggregation strategy in modern FL, Federated Averaging (FedAvg) [29], averages the uploaded clients' model as the updated server model. Mathematically, the aggregation is represented as $w^{t+1} = w^t - \eta \sum_{i \in S_t} \frac{|D_i|}{n} \Delta w_k^t$ [23]. Because FedAVG is only capable of handling Non-IID data to a limited degree, current FL studies proposed improvements in either local training or global aggregation based on it.

**FedProx [25]** FedProx improves local training by directly adding a $L_2$ regularization term, $\mu$, $\frac{\mu}{2}||w - w^t||^2$ controlled by hyperparameter $\mu$, in the local objection function to shorten the distance between the server and the client distance. Namely, this regularization enforces the updated model to be as close to the global optima as possible during aggregation. In our experiment, we carefully tuned $\mu$ to achieve the current results.

**FedNova [38]** FedNova aims to tackle imbalances in the aggregation stage caused by different levels of training (e.g., a gap in local steps between different clients) before updating from different clients. The idea is to make larger local updates for clients with deep level of local training (e.g.,

a large local epoch). This way, FedNova scales and normalizes the clients' model before sending them to the global model. Specifically, it improves its objective from FedAvg to $w^{t+1} = w^t - \eta \frac{\sum_{i \in S_t} |D^i| \tau_i}{n} \sum_{i \in S_t} \frac{|D^i| \Delta w_k^t}{n \tau_i}$ [23].

**Scaffold [18]**   Scaffold introduces variance reduction techniques to correct the 'clients drift' caused by gradient dissimilarity. Specifically, the variance on the server side is represented as $v$, and on the clients' side is represented as $v_i$. The local control variant is then added as $v_i - v + \frac{1}{\tau_i \eta}(w^t - w_i^t)$. At the same time, the Scaffold adds the drift on the client side as $w^t = w^t - \eta(\Delta(w_t; b) - v_i^t + v)$ [23].

**Virtual Homogeneous Learning (VHL) [37]**   VHL proposes to calibrate local feature learning by adding a regularization term with global anchor for local training objectives $\mathbb{E}_{(x,y) \sim P_k} l(\rho \circ \psi(x), y) + \mathbb{E}_{(x,y) \sim P_v} l(\rho \circ \psi(x), y) + \lambda \mathbb{E}_y d(P_k(\psi(x)|y), P_c(\psi(x)|y))$. They theoretically and empirically show that adding the term can improve the FL performance. In the implementation, they use untrained StyleGAN [19] to generate global anchor data and leave it unchanged during training.

A comprehensive experimental study of FL can be found here [23]. Also, a survey of heterogeneous FL is here [48].