# OpenReview forum: "Federated Virtual Learning on Heterogeneous Data with Local-global Distillation"
_NeurIPS.cc/2023/Conference — Submitted to NeurIPS 2023_

### Official Review · Reviewer_TJpJ · 2023-07-01

**Soundness:** 2 fair
**Presentation:** 2 fair
**Contribution:** 2 fair
**Rating:** 6
**Confidence:** 4

**Summary:**

This paper proposes a method called FedLGD that utilizes distilled virtual data on both clients and the server to train FL models. To address the synchronization issue and class imbalance, the authors use iterative distribution matching to distill the same amount of local virtual data on the clients for local model training, thereby improving the efficiency and scalability of FL. The authors also reveal that training on local virtual data exacerbates the heterogeneity issue in FL. To address this problem, they use federated gradient matching to distill global data on the server and add a regularization term to the local loss function to promote the similarity between local and global features. They evaluate the proposed FedLGD method on benchmark and real datasets and show that FedLGD outperforms existing heterogeneous FL algorithms.

**Strengths:**

1. The authors use visualization to reveal the limitation of local data distillation in federated virtual learning, which makes the motivation of the proposed method clear.
2. The proposed method preserves local data privacy by leveraging averaged local gradients to distil global virtual data.
3. The experiment results validate performance improvement and privacy protection.


**Weaknesses:**

1. The initialization for data distillation requires each client to calculate the data statistics and the server to aggregate these statistics, which still raises privacy concerns since the statistics contain some private information. How about using random initialization or other strategies? The authors need to justify it.
2. Compared with VHL that uses untrained StyleGAN without further updates, the proposed FedLGD method needs to update the global virtual data iteratively.  Therefore, it is not surprising that FedLGD outperforms VHL. If the StyleGAN can be updated the same number of times as FedLGD, does FedLGD still outperform it? This requires justification or experimental validation.
3. The structure of the proposed method is not clear enough, which makes it difficult to follow. The authors first present the overall pipeline and then describe each component. However, the connection between the components and the overall pipeline is not clear. This requires significant revision.
4. The presentation quality is not satisfactory. There are too many typos and grammatical errors. Some notations are unclear, e.g., $i$ represents both the data index and client index; the subscript $t$ disappears in many places; $\tau$ is a set but denoted as a scalar in the caption of Figure 2.


**Questions:**

1. For the selected rounds, is the regularization term $L_{con}$ needed for the local model update? As shown in Algorithm 1, the term disappears in these rounds.
2. In Table 1, why is VHL better than the proposed method on SVHN with IPC=10?


**Limitations:**

1. As pointed out by the authors, data distillation incurs additional communication and computation cost. Further investigation is required to enhance the efficiency.
2. The proposed method performs well but lacks of theoretical analysis to support its performance improvement.

---

> ### Author Rebuttal · Authors · 2023-08-10
>
> > Privacy concern on sharing the local dataset statistics
>
> Thank you for the comments. However, we would like to point out that FedLGD aims to reduce local privacy leakage *by training and sharing gradients w.r.t. local virtual data*. The global virtual data is designed for *regularizing local training on the feature extractors*, so that we can handle the feature heterogeneity issue among clients. We had reported using different initialization strategies for initializing both local and global virtual data on DIGITS datasets in Appendix C.7 (Table 9). Furthermore, we provide an alternative initialization that does not require sharing local data mean and variance for global data distillation in our rebuttal and report the consistent good results in rebuttal PDF (Table 1). The performances are almost identical for global virtual data to be initialized with either random noise or local statistics, so sharing local statistics to the server is in fact optional. We will add the discussion to the Appendix of our revised version.
>
> We have provided more comprehensive analysis on the positive impacts of using FedLGD for privacy preservation in the **Justification of privacy** section of the general response.
>
> > Justification on iteratively updating global virtual data v.s. StyleGAN
>
> We would like to point out that training StyleGAN for generating global virtual data in FL is a challenging task: First, it requires either a large in-domain global dataset or an algorithm that incorporates all clients’ data. The latter part is itself an outstanding research topic which is beyond the scope of this paper. Second, updating GAN requires clients and the server to host an additional StyleGAN model, whereas we rely on classification models only. Thus, enabling global virtual data to be updated through global iterations is a key contribution of FedLGD.
>
> > Explanation of the structure of Sec. 3
>
> Thanks for the great comments on paper writing, we tried to put the notations in the same paragraph for easy reference and provided a notation table in Appendix A in our original submission. For the structure of Sec. 3, we had several internal discussions and carefully chosen the current version for presentation. For example, we want to introduce the fundamental part of federated virtual learning, so we began with the introduction of local virtual data. Then, we introduced the heterogeneity problem caused by FVL, so we brought in the concept of global virtual data to formulate the local training objective of FedLGD (Eq. 3). In the last sentence of the paragraph, we pointed out the question we wanted to solve with our novel design of federated gradient matching (“At this point, a critical problem arises: What global virtual data shall we use?”). We will carefully read through and revise the paragraph to make it easier to follow.
>
> > Typos and grammar errors
>
> Thank you for pointing out our typos. We have corrected the caption in Fig. 2. We agreed that our paper had a lot of notations since we needed to cover a variety of variables for explaining our pipeline. We tried our best to carefully assign consistent notations for each variable and provided a notation table in the Appendix A. We will perform additional rounds of proof-reading for revising our notations.
>
> > Explanation of $L_{Con}$ in Algorithm 1
>
> Thank you for pointing out the subtle design of FedLGD. No, we would not add the regularization term for local training in the selected rounds because we required the averaged gradients w.r.t. *only* the cross-entropy loss from clients. Therefore, we disabled the regularization term in the selected iterations for synthesizing global virtual data.
>
> > Experimental results in Table 1
>
> Thanks for your careful review. In the majority of cases, and as substantiated by the experimental results, our method demonstrates superior performance to VHL, with a significant margin exceeding 2%. Indeed in two specific instances identified by reviewer 5yts, VHL exhibits better results, albeit by relatively narrow margin: SVHN+ConvNet (0.4%). This discrepancy may be attributed to the nature of SVHN, which are three-channel (RGB) images. As a result, the anchors generated by the styleGAN in VHL might more effectively capture the RGB information. Yet, it is important to emphasize that under the same conditions, our method exhibits much better results compared to VHL for the other clients.
>
> > Theoretical analysis of FedLGD
>
> Please see the **Theoretical analysis of FedLGD** section in the general response.
>
> [r1] Dataset distillation: A comprehensive review.

---

> > ### Comment · Reviewer_TJpJ · 2023-08-14
> >
> > The authors have resolved some of my main concerns (especially the privacy and initialization), and thus I have increased my rating.

---

> > > ### Author Response · Authors · 2023-08-14
> > > **Appreciate your feedback on our responses**
> > >
> > > We are pleased to learn that our explanations and clarifications have been positively acknowledged and your major concerns regarding privacy and initialization are well-dressed in our replies. Your comments and feedbacks have been immensely valuable. We would like to express our sincere appreciation once more for your time and the dedication for reviewing our work.

---

### Official Review · Reviewer_zf9v · 2023-07-05

**Soundness:** 3 good
**Presentation:** 3 good
**Contribution:** 3 good
**Rating:** 6
**Confidence:** 3

**Summary:**

This work proposes a method to address data heterogeneity from the perspective of dataset distillation, named FedLGD. Specifically, the proposed iterative distribution matching and federated gradient matching strategies are used to iteratively update the local balanced data and the global shared virtual data, and the global virtual regularization is applied to coordinate the data domain drift between clients effectively. This method can effectively solve the problem of client data imbalance and domain drift in heterogeneous data scenarios. Extensive experimental results show the effectiveness of the proposed method.

**Strengths:**

1. The problem of data heterogeneity studied in this paper is important for applying federated learning in real-world scenarios.
2. The idea of this paper to solve the problem of data heterogeneity in federated learning through the dataset distillation method is novel.
3. The authors perform various experiments to analyze the proposed method


**Weaknesses:**

1. Some symbols are written differently. The authors should unify these symbols. In section 3.1, local virtual data is written as $\widetilde{D}_{I}$, but in section 3.2 and 3.3 is written as $\widetilde^{D}{c}$, in Figure 2 is $\widetilde{D}_{t}^{c_{I}}$. In Eq. (3), $L_ {Con} $is about \ widetilde ^ {D} {g} and \ widetilde {D} ^ {c} function. But they do not appear in Eq. (4). The authors should give more details about Eq. (3) and (4).
2. other approaches to address heterogeneity are personalized federated learning, e.g. FedAMP[1] Ditto[2] KT-pFL[3], etc. However, it is not mentioned in related work, and there is no comparison in experimental methods.
3. The legend and curve in Figure 3a do not match. In Table 1, ResNet18 generally performs worse than CNN. The authors mention that overfitting may be happening (at line 275). The authors should increase the dataset size or use a smaller model to make the results more convincing.
4. According to the results in Figure 4, the visualization results of FedLGD do not define the boundaries of each class well. Although FIG. 4 can prove that FedLGD solves the data drift of both clients, the degree of clustering of each class looks reduced. The authors should analyze it further.


[1] Huang, Yutao, et al. "Personalized cross-silo federated learning on non-iid data." Proceedings of the AAAI Conference on Artificial Intelligence. Vol. 35. No. 9. 2021.
[2] Li, Tian, et al. "Ditto: Fair and robust federated learning through personalization." International Conference on Machine Learning. PMLR, 2021.
[3] Zhang, Jie, et al. "Parameterized knowledge transfer for personalized federated learning." Advances in Neural Information Processing Systems 34 (2021): 10092-10104.


**Questions:**

1. Can the proposed approach be generalized to other data types such as text and corresponding text models?
2. Can the proposed method be generalized to more client scenarios which is more practical in reality? Is the proposed method theoretically guaranteed?




**Limitations:**

The authors discuss the limitations in section 5.

---

> ### Author Rebuttal · Authors · 2023-08-10
>
> > Explanation of the notations and symbols
>
> In Sec. 3.1, we started with the classical FL setting and derived to FVL setting, so we used $\widetilde{D}\_{i}$ to represent each client’s virtual data. Beginning Sec. 3.2, we introduced the global and virtual data, so we used $\widetilde{D}^c_i$ and $\widetilde{D}^g_i$ instead. We thank the reviewer for the careful review in Fig. 2 and we will correct the notation to $\widetilde{D}^c$.
> Regarding Eq. 3 & 4, we explained how $L_{Con}$ is related to $\widetilde{D}^g$ and $\widetilde{D}^c$ in line 201-202. We apologize for the confusion, and we will add the detailed explanation of Eq. 3.
>
> > Justification on the differences between FedLGD and PFL
>
> Thanks for pointing out the PFL literature. This work focuses on the *classical FL setting*, where the objective is to optimize a single global model that generalizes well to heterogeneous clients, as stated in lines 131-132. Specifically, we aim to evaluate a single global model on the global distribution (a joint of local distributions), a concept often referred to as out-of-distribution (OOD) testing in the literature, such as [r1]. Conversely, PFL seeks to test various local models on their individual testing sets, an approach known as in-distribution testing. Given the differing goal and evaluation criteria, we chose not to include PFL in our comparison. Nonetheless, in response to your suggestion, we have added a discussion on PFL in our related work section.
>
> [r1]Motley: Benchmarking heterogeneity and personalization in federated learning.
>
> > Explanation of overfitting and Fig. 3a
>
> We’d like to affirmatively confirm that the statements to the figure are correct. Line 275 is to explain why ConvNet performs better than ResNet. We meant “overfitting” as the phenomenon that when we use small, distilled virtual images as the training data to train heavy architectures, the model tends to overfit (Please check Fig. 1 in the rebuttal PDF). As the reviewer suggested, we indeed showed that using more virtual data (larger IPC) and a smaller model (such as ConvNet) resulted in higher accuracy in Table 1. Notably, the overfitting observation was consistent with other data distillation literature [*45*] as we stated in line 274.
>
> We want to emphasize that the visualization in Fig. 3a is for a totally different purpose – showing our proposed contrastive regularization loss performs better than MMD, in which we plotted the averaged accuracies of DIGITS and CIFAR10C separately on the given IPC and model, while varying the $\lambda$ of the different choices for regularization terms.
>
> > Explanation of Fig. 4
>
> We plotted the tSNE figure on the *features layer that distribution loss was added on* to show whether our regularization term could group the features from different clients together if they’re from the same class. Thus, the main purpose is not to have a clear boundary in this feature space but to check the grouping effect. We add a further layer for classification, which helps increase the separability.
>
> > The generalization of FedLGD on text data and models
>
> We believe that it is promising to generalize our methods to text data and NLP models, given recent studies have explored data distillation on text data [r1,r2,r3]. However, the data distillation strategies for text data and NLP models are slightly different from those for image data and CNNs, thus we believe modifications are needed in virtual data generation procedure. We believe our novel ideas and successful demonstration on heterogeneous images can inspire the community to explore more on the NLP area in the near future.
>
> [r1]Data distillation for text classification.
>
> [r2]Dataset Distillation with Attention Labels for Fine-tuning BERT.
>
> [r3]Soft-label dataset distillation and text dataset distillation.
>
> > Generalization of FedLGD on more client scenarios
>
> We thank the reviewer for the question. We conducted experiments on CIFAR10C dataset to show that FedLGD consistently performed well when there were 57 heterogeneous clients. In general, the computation cost and communication overhead will not increase if the number of clients increases, so we believe our proposed method is scalable to the number of clients.
>
> > Theoretical analysis of FedLGD
>
> Please see the **Theoretical analysis of FedLGD** section in the general response.
>
>
> [r1]Dataset distillation: A comprehensive review.

---

> > ### Author Response · Authors · 2023-08-16
> > **Thank you for your comments and look forward to further discussions**
> >
> > Dear reviewer zf9v,
> >
> > We appreciate the feedback and suggestions you've provided to our paper. We've put in our utmost effort to thoughtfully address these points and have included further details of privacy and theoretical analysis in the overall response. We sincerely hope our responses have solved your concerns to our paper, and we remain fully available to respond to any extra inquiries that may arise during the discussion phase.
> >
> > Best Regards,
> > FedLGD authors

---

### Official Review · Reviewer_DZFG · 2023-07-07

**Soundness:** 2 fair
**Presentation:** 3 good
**Contribution:** 2 fair
**Rating:** 4
**Confidence:** 4

**Summary:**

To solve the challenges of synchronization, efficiency, and privacy, this paper presents a local-global distillation mechanism for FL (FedLGD). In FedLGD, an iterative distribution matching scheme is proposed to distill global virtual data to alleviate the heterogeneous problem. Experiments have shown superiority of FedLGD compared with existing FL methods.

**Strengths:**

1. The whole pipeline of FedLGD is well depicted in Figure 2. Each component involved in the pipeline is carefully designed.

2. It is an interesting idea to solve the existing FL challenges from the virtual learning perspective. This can inspire future studies in this direction.

3. Experimental results look solid with sufficient implementation details.


**Weaknesses:**

1. It seems that only feature heterogeneity is considered in this work. How the proposed method performs on different heterogeneous cases should be discussed.

2. The definition of small distilled dataset is not very clear, which can affect the readers’ understanding towards the motivation and detailed technical parts.

3. Privacy concern. Since there are image-level data transferred between the server and clients, it is better to further discuss the potential privacy-preserving risks.


**Questions:**

What is the detailed setting of Figure 1? Is it reasonable to use the size of the overlapped area to represent the level of heterogeneity? Is there any theoretical basis? Also, will the results be different when using different synthetic schemes?


**Limitations:**

Yes. The authors have addressed the limitations.

---

> ### Author Rebuttal · Authors · 2023-08-10
>
> > Justification of the client heterogeneity considered in FedLGD
>
> We studied on both *label and feature shift* among clients as we stated in the last paragraph of our Introduction(line 64-76). Intuitively, generating the same Images Per Class (IPC) can balance label shift. Particularly, we showed FedLGD could handle label shifts by our CIFAR10C experiments where the local datasets are sampled with Dirichlet distribution. The effectiveness of FedLGD in mitigating feature shifts was shown in the experiments in DIGITS, CIFAR10C, and RETINA datasets.
>
> > Explanation of the definition of small distilled dataset
>
> We tried to explain our FVL setting in line 134-140, where a much smaller virtual data is used for local training in FL for synchronization and efficiency of the FL system, which follows the common purpose of data distillation literature, including the ones in FL [*10, 16, 40*]. Here, the ‘small’ was referred to as a comparison to the size of local real datasets (in line 138), and we empirically followed the original data distillation papers to set the numbers of images per class to 10 and 50 [*45, 46*]. The word ‘distilled’ meant the *method* for synthesizing local and global virtual data as described in Sec. 3. We referred to ‘distilled data’ as ‘virtual data’ in alignment with a closely related work, VHL.
>
> > Justification of the privacy concern on sharing global virtual data
>
> Thanks for the great suggestion. We indeed carefully thought about privacy concerns in our original submission.
> We first want to clarify that the ‘image-level’ information pointed by the reviewer is actually inverted from the shared global model update, which is accessible to every client by default in classical FL such as FedAvg. We do NOT directly share local private images or locally generated images as other literature does in CVPR2023 [*40*].  Instead, we shared local data mean and variance to global server for global server distillation. Except for local data statistics, we shared the *SAME* level of information to the server as in FedAvg. We also showed that FedLGD can preserve higher privacy against Membership Inference Attack compared to FedAvg in Appendix D.6. In addition, we provide an alternative initialization that does not require sharing local data mean and variance for global data distillation in our rebuttal and report the consistent good results in rebuttal PDF (Table 1). Lastly, in line 345, we stated that “Our future direction will be investigating privacy-preserving data generation” to further enhance FedLGD’s privacy preservation.
>
> > Explanation of the feature heterogeneity shown in Fig. 1
>
> We directly fed in the randomly sampled raw data from the two datasets for plotting Fig. 1. We followed the visualization of VHL to inspect the data heterogeneity by tSNE plot [*37*] (as depicted in line 39-42 in our original submission). tSNE is a statistical method for visualizing high-dimensional data by giving each datapoint a location in a two or three-dimensional map [r1]. It could be observed that after distillation, the data distribution of client 0 became more clustered and separated from the distribution of the data distribution of client 1. In addition, we statistically show the distance between real and virtual data in the rebuttal PDF (Table 2).
>
> [r1] Visualizing data using t-SNE.

---

> > ### Author Response · Authors · 2023-08-16
> > **Thank you for your comments and look forward to further discussions**
> >
> > Dear reviewer DZFG,
> >
> > We appreciate the feedback and suggestions you've provided to our paper. We've put in our utmost effort to thoughtfully address these points and have included further details of privacy and theoretical analysis in the overall response. We sincerely hope our responses have solved your concerns to our paper, and we remain fully available to respond to any extra inquiries that may arise during the discussion phase.
> >
> > Best Regards,
> > FedLGD authors

---

> ### Author Response · Authors · 2023-08-20
> **Appreciate your feedback on our responses**
>
> Dear Reviewer DZFG,
>
> As the Discussion stage is about to end, we sincerely look forward to knowing whether our responses have addressed your initial questions. We are more than happy to answer your remaining concern and appreciate your inputs and feedback very much. Thank you!
>
> Best Regards, FedLGD Authors

---

### Official Review · Reviewer_5ytS · 2023-07-21

**Soundness:** 3 good
**Presentation:** 3 good
**Contribution:** 3 good
**Rating:** 5
**Confidence:** 3

**Summary:**

This paper introduces an approach on Federated learning using dataset distillation techniques

**Strengths:**

1. The idea of using dataset distillation for FL is interesting
2. The solution is reasonable
3. The experimental results show the effectiveness of the proposed approach

**Weaknesses:**

1. In a few equations, the details is not provided. For instance $L_CE$ in 3, $Dist$ in 5. The paper should be self-contained
2. The technical contribution is low
3. In the experimental results, Table 1, can you highligh both first and second place? In MNIST-M, the winner should be VHL/R, 85.7> FedLGD 85.2.

**Questions:**

1. Can you try to provide details on those equations, see above?
2. I didn't see how the sizes of different clients play a role in the global data distillation. Can you provide some details?
    Also, what is the overall objective functions for both local training and global training? The current equations  are not very clear on this.

**Limitations:**

The limitation on using virtual data should be discussed. Is there any drawbacks on using fake data instead of real data?

---

> ### Author Rebuttal · Authors · 2023-08-10
>
> > Explanation of Eq. 3 and Eq. 5
>
> We stated that $L_{CE}$ is the cross-entropy loss in line 197 and $L_{Dist}$ is defined the same as that in [*46*] in line 219. Apologize for the confusion, following your suggestion, we have added the detailed equation of $L_{CE}$ and $L_{Dist}$ in our revised version.
>
> > Justification of our technical contribution
>
> We would like to take the chance to re-emphasize our technical contribution in the following:
> * We applied data distillation to FL pipeline to handle label shift, asynchronization problems in FL, and we were the first to point out data distillation may amplify the data heterogeneity among clients.
> * We proposed Iterative Distribution Matching to iteratively inpaint the global information to local virtual data using the up-to-date global model.
> * We proposed Federated Gradient Matching to efficiently incorporate the bi-level optimization problem of data distillation using gradient matching into the classical FL pipeline. The virtual global data could be used to regularize feature heterogeneity among clients.
> * Through comprehensive experiments on benchmark and real-world datasets, we showed that FedLGD outperformed existing state-of-the-art FL algorithms.
>
> > Experimental results in Table 1
>
> Thanks for your careful review. In the majority of cases, and as substantiated by the experimental results, our method demonstrates superior performance to VHL, with a significant margin exceeding 2%. Indeed in two specific instances identified by reviewer 5yts, VHL exhibits better results, albeit by relatively narrow margins: SVHN+ConvNet (0.4%) and MNIST-M+ResNet (0.5%). This discrepancy may be attributed to the nature of SVHN and MNIST-M, which are three-channel (RGB) images. As a result, the anchors generated by the styleGAN in VHL might more effectively capture the RGB information. Yet, it is important to emphasize that under the same conditions, our method exhibits much better results compared to VHL in all other clients. Following the valuable suggestion, we update the bold highlighting in our Table 1.
>
> >  Explanation of the effect of the size of different clients on global data distillation
>
> We thank the reviewer for the clarification question. To make sure we better answer your first question, we assume two possible scenarios about the *size*: 1) the number of local real data and 2) the number of selected clients in each round of training. For 1), since we distilled the same amount of local virtual data for each client, namely $|\tilde{D}^c_i|$ = $|\tilde{D}^c_j|$ for i, j $\in$ [N] (see line 162-163), the number of local real data would not directly affect global data distillation. For 2), we had investigated a different number of selected clients in each round of training and the results were presented in Table 2. We observed consistent performance w.r.t. different number of selected clients, which indicate a stable results on global data distillation, since we used *averaged* gradients  for updating global virtual data (Eq. 5).
>
> Regarding the reviewer’s second question, we would like to first re-state our overall training pipeline:
>
> 1.Initialization local virtual data
>
> 2.if selected iterations:
>
>     Clients: update local virtual data and model;
>
>     Server: update global virtual data and aggregate model;
>
> 3.else:
>
>     Clients: update model;
>
>     Server: aggregate model;
>
> We would like to clarify that we had two types of local training — 1) local model updating and 2) local virtual data distillation. The objective of 1) was depicted in Eq. 3 where we used cross-entropy loss and contrastive loss to train local models. The objective of 2) was depicted in Eq. 2 where we updated local virtual data with the MMD loss and the up-to-date global model.
> Note that we didn’t train the global model with gradient descent (instead we performed FedAvg aggregation), so we referred to “global training” in the question as global data distillation. We updated global virtual data with the gradient matching as depicted in Eq.5 where we forced the global gradients to match the averaged local gradients.
>
> > Explanation on the limitations of FedLGD
>
> In the conclusion of our original submission, we pointed out the additional communication overhead and computational cost in data distillation. Note that if we train FL for long iterations, FedLGD still could be more efficient compared with training on real data as we analyze in Appendix C.3. Additionally, like the phenomenon that happens in existing Data Distillation literatures, models trained on virtual data may perform worse than training on real data. While in FedLGD, there is a trade-off on such limitation and enjoying the benefits of using virtual data in FL, including improving model training efficiency, FL synchronization and stronger inversion defence on the model parameters.

---

> > ### Author Response · Authors · 2023-08-16
> > **Thank you for your comments and look forward to further discussions**
> >
> > Dear reviewer 5ytS,
> >
> > We appreciate the feedback and suggestions you've provided to our paper. We've put in our utmost effort to thoughtfully address these points and have included further details of privacy and theoretical analysis in the overall response. We sincerely hope our responses have solved your concerns to our paper, and we remain fully available to respond to any extra inquiries that may arise during the discussion phase.
> >
> > Best Regards,
> > FedLGD authors

---

> > ### Comment · Reviewer_5ytS · 2023-08-16
> > **thanks for the clarification**
> >
> > The authors have addressed some of my concerns; I will increase my score.

---

### Author Rebuttal · Authors · 2023-08-10

> Appreciation to the reviewers

We thank the reviewer for the positive comments about our FedLGD design(5ytS, DZFG, zf9v), FedLGD's effectiveness through comprehensive experiments(5ytS, DZFG, zf9v, TJpJ), the presentation of FedLGD (DZFG), and the privacy-preservation mechanism of FedLGD(TJpJ).
We would like to take the chance to re-emphasize our technical contribution in the following:
* In this work, we focused on Federated Virtual Learning (FVL), a new FL framework and promising approach to apply data distillation-based synthetic data to FL pipeline to handle label shift, asynchronization problems in FL. We proposed FedLGD, which incorporated iterative local and global data distillation to achieve good performance with limited amounts of distilled virtual data.  We proposed Iterative Distribution Matching to inpaint the global information to local virtual data using the up-to-date global model. Through local virtual data distillation, *class-balanced* synthetic data were generated to facilitate FL training.
* We proposed Federated Gradient Matching to efficiently incorporate the bi-level optimization problem of data distillation using gradient matching into the classical FL pipeline. The virtual global data could be used to *regularize feature heterogeneity* among clients.
* Through comprehensive experiments on benchmark and real-world datasets under various settings, we showed that FedLGD outperforms the existing state-of-the-art FL algorithms.

We have carefully addressed the comments from the reviewers. We sincerely appreciate it if reviewers could reconsider your rating of our paper. We are delighted to further discuss with you if any further questions arise during the discussion phase. Thank you again for your invaluable time and suggestions.

> Justification of privacy

First, we would like to point out that FedLGD aims to reduce local privacy leakage *by training and sharing gradients w.r.t. local virtual data*. The global virtual data is designed for *regularizing local training on the feature extractors*, so that we can handle the feature heterogeneity issue among clients.
Second, we want to clarify that the shared 'image-level' information is actually inverted from the shared global model update, which is accessible to every client by default in classical FL such as FedAvg. We do NOT directly share local private images or locally generated images as other literature does in [*40*].  Instead, we shared local data mean and variance to global server for global server distillation. Except for local data statistics, we shared the *SAME* level of information to the server as in FedAvg. We also showed that FedLGD can preserve higher privacy against Membership Inference Attack compared to FedAvg in Appendix D.6.
In addition, we provide an alternative initialization that does not require sharing local data mean and variance for global data distillation in our rebuttal and report the consistent good results in rebuttal PDF (Table 1).
Lastly, in line 345, we stated that "Our future direction will be investigating privacy-preserving data generation" to further enhance FedLGD's privacy preservation.

> Theoretical analysis of FedLGD

Yes, we can show theoretical insights on FedLGD, which mainly relies on the existing tools [*37*, r1].

First, the generalization performance of model $f$ on distribution $P(x,y)$ can be analyzed using statistical margin (SM) following [*37*]. Here $f$ is optimized using FedLGD with minimizing Eq. (3). Let $f=  \phi \circ h $ be a neural network composed of a feature extractor $\phi$ and a classifier $h$. Since we obtain local virtual data $\widetilde{D}_c$ via distribution matching with real data, thus we assume $P_{\widetilde{D}} \approx P$. Similar to VHL[*37*] (Theorem A.2), we have the lower bound of SM for FedLGD as: $$ E_{h \leftarrow P_g} SM(h, P)
\geq E_{h \leftarrow P_g} SM(h, \widetilde{D})-E_{h \leftarrow P_g}[SM(h, P_g)-SM(h, \widetilde{D})] |-E_y d(P(\phi | y), P_{g}(\phi | y)),$$
where $P_g$ denote global virtual data distribution.

The key distinction between FedLGD and VHL is the last term, also known as distribution matching objective between $P$ and $P_g$. For maximizing SM to achieve strong generalization, we want to show SM has a tight lower bound. Therefore, upper-bounding the last term is desired. Note that in VHL, the global virtual data is generated from an un-pretrained StyleGAN. In contrast, our approach employs the *gradient matching* strategy to synthesize the global virtual data. Due to the complexity of data distillation steps, we consider the analysis on a kernel ridge regression model with a random feature extractor. Based on the Proposition 2 of [r1], the first-order distribution matching objective (the last term) is approximately equal to gradient matching of each class (i.e., $\mathcal{L}_{Dist}$ (Eq 5)) . Namely, minimizing Eq 5 in FedLGD implies minimizing the last term in that setting. Hence, using gradient matching generated global virtual data elicits the model's SM a tight lower bound and proved generalization.

[r1] Dataset distillation: A comprehensive review.

Note: We denote the new references used in rebuttal as [r#].

---

### Decision · Program_Chairs · 2023-09-21

**Decision:**

Reject

**Comment:**

The paper proposes a local and global distillation scheme to mitigate feature and label imbalances in heterogeneous federated learning. Reviewers raised several concerns, and the rebuttal did address a few of those, moving the score in a positive direction. However, there was a lack of sufficient excitement, and the paper fell right on the border. The area chair had to review the paper personally. The idea of local and global knowledge distillation significantly complicates the federated learning setting. While targeted distillation can solve and mitigate some problems, it is likely to introduce several complications such as privacy concerns, efficiency issues, and more in the federated setting. There are too many variables to consider, such as how much distillation and how many rounds of distillation. All of these factors can significantly impact the benchmarks.

The proposals seem to rely heavily on manual engineering to make them work. The datasets under consideration are also simple and small. The authors should argue and provide more rigorous evaluations of different components of their method and how they interact with each other. It's important to understand how much global distillation versus local distillation helps individually. This is an empirical paper, and there should be more discussion on when this complex double distillation process could potentially worsen the situations. It falls short of the publication standards at NeurIPS, which require a thorough evaluation of a pipeline with many complex components.